# Precipitation projections using a spatiotemporal distributed method: a case study in the Poyang Lake Watershed based on MRI-CGCM3

Ling Zhang [1], Xiaoling Chen[1, 2], Jianzhong Lu[1, *], Dong Liang[1]

[1]State Key Laboratory of Information Engineering in Surveying, Mapping and Remote Sensing, Wuhan University, Wuhan 430079, China

[2]Key Laboratory of Poyang Lake Wetland and Watershed Research, Ministry of Education, Jiangxi Normal University, Nanchang 330022, China

* *Correspondence to*: Jianzhong Lu (lujzhong@whu.edu.cn)

**Abstract.** To bridge the gap between large-scale GCM (Global Climate Model) outputs and regional-scale climate requirements of hydrological models, a spatiotemporally distributed downscaling model (STDDM) was developed. The STDDM was done in three stage: (1) upsampling grid-observations and GCM (Global Climate Model) simulations to spatially continuous finer−grids; (2) creating the mapping relationship between the observations and the simulations, differently in space and time; (3) correcting the simulation and produced downscaled data in spatially continuous grid scale. We applied the STDDM to precipitation downscaling in Poyang Lake Watershed using MRI-CGCM3 (Meteorological Research Institute Coupled Ocean-Atmosphere General Circulation Model3), with an accepted uncertainty of $\leq 4.9\%$; then created future precipitation changes from 1998 to 2100 (1998-2012 in the historical and 2013-2100 in the RCP8.5 scenario). The precipitation changes increased heterogeneities in temporal and spatial distribution under future climate warming. In terms of temporal patterns, the wet season become wetter while the dry season become drier. The frequency of extreme precipitation increased while that of the moderate precipitation decreased. Total precipitation increased while rain days decreased. The max continuous dry days and the max daily precipitation both increased. In terms of spatial patterns, the dry area exhibited a drier condition during the dry season; the wet area exhibited a wetter condition during the wet season. Analysis with temperature increment showed precipitation changes can be significantly explained by climate warming, with $p < 0.05$ and $R \geq 0.56$. The precipitation changes and explains indicated the downscaling method is reasonable and the STDDM could be applied in the basin-scale region based on a GCM successfully. The results implicated an increasing risk of flood-droughts under global warming, which were a reference for water balance analysis and water resource planting.

## 1 Introduction

Global warming has caused temporal and spatial redistributions of precipitation (Frei et al. 1998; Trenberth et al. 2011) and has increased the frequency and intensity of floods and droughts, seriously threating social systems and ecosystems (Pall et al, 2000; Dai, 2013). To the fragile ecological and living environments, what the future hydrological situation will be under future global warming is a crucial question to avoid or reduce damages from climate warming.

Global Climate Models (GCMs) are basic tools for assessing the effects of future climate change and provide an initial source for future climates (Xu, 1999). However, GCMs have coarse global resolutions ranging from $1° \times 1°$ to $4° \times 4°$, and are not applicable in regional scales, such as watersheds. Downscaling algorithms have been developed to link the global-scale GCM outputs and the regional-scale climate variables, including dynamic (Giorgi, 1990; Teutschbein and Seibert, 2012) and statistic (Wilby et al., 2007; Chu et al., 2010) models. The dynamic method employs regional climate models (RCMs) that are nested inside GCMs based on the complex physics of atmospheric processes and involves high computational costs. Limited by an insufficient understanding of the physical mechanism and expensively computing resources, the dynamic downscaling model cannot easily satisfy small and mid-size region as the Poyang Lake Watershed. Unlike dynamic downscaling, statistic downscaling constructs an empirical relationship between climate variables of the global-scale and local-scale, with inexpensive computations. Benefiting from inexpensive computations and easy implementations, downscaling methods have been widely used, including regression models (Labraga et al. 2010, Quintana et al. 2010; Zorita et al. 1999), weather typing schemes (Boéj et al. 2007; ENKE et al. 2005) and weather generators (Mullan et al., 2016; Baigorria and Jones et al., 2011).

Most statistical downscaling methods are conducted on discrete stations (Charles et al., 1999; Zhang et al., 2005; Maurer et al., 2008; Mullan et al., 2016; Alaya et al., 2017; Chen et al., 2018) and produce downscaled data the in the station scale., including single-station and multi-station methods. The single-station method produces the downscaled climate variable at a single point (or watershed average), or independently at several points (Zhang et al., 2005; Maurer et al., 2008). The multi-station method generates the downscaled climate variable dependently for multiple sites (Charles et al., 1999; Alaya et al., 2017; Chen et al., 2018). For both the single-station and multi-station methods, the specific downscaling relationship and downscaled climate variable are both discrete in the station scale, instead of being spatially continuous in a grid-scale of a

finer-resolution. Compared to the spatially continuous grid data, discrete stations are sparse. As underlays of local region are
complex with different topographies, land covers, and clouds coverage, the discrete point-scale data underrepresents the spatial
variability. For ungauged areas without station coverage, it is inviable to obtain high-quality downscaling relationships and
downscaled local climate variables. Moreover, compared to point-scale data, spatially continuous grid data can express the
spatial distribution of climate variables more accurately and clearly; thus express the spatial correlation and heterogeneity
more accurately and clearly. Additionally, spatially continuous grid data can be directly used in a spatially distributed or semi-
distributed hydrological model, such as Crest (Wang et al., 2011), VIC (Lohmann et al. 1998), and MIKE SHE (DHI, 2014),
which is the forefront of international hydrological scientific research (Beven et al. 1990). Spatially continuous downscaled
climate data can also be easily integrated with remote sensing data of geologies, topographies, soils, or land covers. In fact,
spatially continuous data is widely used in the rapidly developing field of remote sensing, which benefits hydrological models
by providing a data source (Engman et al., 1991). Therefore, the downscaling method processed on spatially continuous data
is of vital importance.
Some downscaling methods could obtain spatially continuous data. Dynamic downscaling methods could produce downscaled
climate variables in spatial continuous grid-scale. However, the downscaled grid-data is commonly limited in the resolution,
coarser than 25 kilometers (Trzaska et al., 2014; Maraun et al, 2010); thus could not be applied to small watersheds. A few
statistical downscaling methods of the weather generator could provide downscaled climate variables in a spatially continuous
scale (Perica et, al., 1996; Venema et al., 2010). The specific algorithms can be divided into three cartographies: transformed
Gaussian processes (Guillot and Lebel, 1999), point process models (Wheater et al., 2005; Cowpertwait et al., 2002), and
spatial-temporal implementation of multifractal cascade models (Lovejoy and Schertzer, 2006). However, few researches have
implicated these approaches on GCM outputs. Furthermore, as the refined data obtain from the weather generator is biased
from the observed data, correction is needed. However, in the researches, there is no observed field of finer-resolution
corresponding to the downscaled scale; thus, not all the spatial unit in the downscaled field could be corrected by the observed
field.
As the factors driving climate variables vary in regions and seasons, the statistical downscaling method should consider the
spatial and temporal heterogeneity (Fowler et al., 2007; Manzanas et al., 2018). Most methods (Charles et al., 1999; Maurer et
al., 2008; Alaya et al., 2017) performed the downscaling for each specific-site (or specific type sites), respectively; thus the
downscaled result showed spatial heterogeneity. However, few downscaling methods consider the spatial heterogeneity in a
spatially continuous scale. In terms of temporal heterogeneity, some downscaling algorithms are processed independently on
months (or seasons) (Boé et al., 2007; Leander and Buishand, 2007). For the different time, the algorithm or parameters are
different; thus the temporal heterogeneity is expressed. However, few downscaling methods consider temporal heterogeneity
combined with spatial heterogeneity in the spatially continuous scale.
To produce downscaled data in a spatially continuous scale and consider temporal heterogeneity combined with spatially
continuous heterogeneity, the study proposed a spatiotemporally distributed downscaling method (STDDM). A finer-
resolution observed field (Hutchinson et al., 1998a; Hutchinson et al., 1998b) is induced as the reference to correct the refined
GCM outputs for each grid and time; subsequently, the corrected data is produced as the downscaled data. The correction is
distributed in time and continuous-space.
The Poyang Lake Watershed is sensitive to climate changes in the East Asian monsoon region and therefore is not immune to
global warming. Redistributions of precipitation due to global warming have resulted in an increased occurrence of extreme
hydrological events, an enhanced flood frequency and intensity (Wang et al., 2009; Guo et al., 2006), a significant decline in
lake level and inundation area (Feng et al. 2012; Zhang et al. 2014), which threatened to fragile wetland and forest ecosystems
(Han et al. 2015, Dyderski et al. 2018), economic developments and human lives (Ye et al., 2011). However, the Poyang Lake
Wetland ecosystem is an internationally important habitat for migratory birds, abundant of biodiversity and regarded as a
Natural Reserve. In addition, the watershed is a commercial grain production area and an important part of the Yangtze River
Economic Belt. As this region is economically and ecologically significant, investigating the future precipitation changes in
the watershed is crucial for protection from climate damages. Previous studies of future precipitation changes in the Poyang
Lake Watershed include temporal and special patterns. Precipitation changes in temporal pattern, focused on intensity and
frequency of precipitation extremes (Hong et al. 2014; Wang et al. 2017), as well as the annual or quarterly total precipitation
(Guo et al., 2010; Guo et al., 2008; Li et al., 2016). In spatial pattern, precipitation change analysis covers five subbasins
(Xinjiang, Raohe, Xiushui, Ganjiang and Fuhe subbasins) (Guo, et al. 2010; Hong, et al. 2014) and 13 discrete meteorological
stations (Li et al. 2016), or 7 coarse grids (Guo, et al. 2008). There has been little research concerning the spatial-temporal
distribution of precipitation in a continual fine-resolution grids space. In addition, driving force analysis of precipitation
changes related to temperatures increment has not been conducted.
In the study, taking Poyang Lake Watershed as a test case, we projected future precipitations based on the spatiotemporally
distributed downscaling method (STDDM), using MRI-GCM3 simulations and meteorological observations. The objects are
as the following: (1) to develop a spatiotemporally distributed downscaling method (STDDM), projecting future climate
variables in spatially continual scale; and (2) to document temporal and spatial changes in precipitation for the Poyang Lake
Watershed in the 21st century and the correlations between these precipitation changes and temperature increment. Future
precipitation changes can provide basic hydrological information necessary to a better understanding of water volumes and
flood-droughts risks; furtherly benefits wetland and forest ecosystem conservation and aids decision-making in development,
utilization, and planning of water resources.
**2 Study area and datasets**
**2.1 Study Area**
Poyang Lake Basin (24°28'-30°05' N and 113°33'-118°29'E) is located in the southeast of China, connected with Yangtze
River in the north (Fig. 1). Within the southeast subtropical monsoon zone, the annual average temperature of the watershed
is 17.5°C. The mean annual precipitation is 1638 mm, with 192 rainy days (daily precipitation $\geq$ 0.1 mm/day) and 173 rain-
free days (daily precipitation < 0.1 mm/day). The rainy season lasts from April to July, occupying about 70% of the annual
total amount. Inter or intra annual precipitation variations are dominated by the southeast and southwest monsoon, mainly in
summer. With a coverage area of 162000 km$^2$, the diversities of topographies also effect on precipitation changes. The
topography varies from high mountains of Luoxiao, Wuyi, and Nanling in east, south and west, with the elevation reaching to
the 2200m, to the depressing of Ji Tai or Ganzhou Depressing in the south or center and alluvial plains of Poyang Lake Plain
in the north, with the elevation reaching to <50 m (1a). The different topography and location generate the uneven distribution
of precipitation in space and produce less rain in the depressing, plains, and hills area because of the leeward sloop, but more
orographic rain in the mountain area for the reason of the windward sloop (1b) (Mingjin et al. 2011). To analyze precipitation
changes in the rich- or poor-rain area, the meteorological stations were classified into dry and wet stations (Fig. 1a and b),
according to the annual precipitation amount. We sorted the annual precipitation averaged over the time from 1961 to 2005,
of the 15 stations. The four stations with the max or min mean annual precipitations are set as dry or wet stations, indicating
the dry or wet area (Fig.1b), respectively.
In the past 50 years of the Poyang Lake Watershed, annual mean temperature indeed experiences a significant (p<0.02) increase
with a change rate of 0.15 °C/10a (Fig.1d), based on the meteorological observations from 1961 to 2005. Under the temperature
increasing condition, the precipitation in temporal and spatial distribution becomes more uneven (Zhan et al. 2011), which
increases the risk of floods and droughts (Li et al. 2016; Ye et al. 2011).
**2.2 Data sets**
Global Climate Models (GCMs) are widely used tools to project future climate change. GCMs from the Coupled Model
Intercomparison Project Phase Five (CMIP5) performs better than other CMIPs such as CMIP3 and CMIP4, with generally
finer resolution and more improved physical mechanism (Sperber, 2013; Taylor et al. 2012). Compared to the other CGMs of
CMIP5, the MRI-CGCM3 (Meteorological Research Institute Coupled Ocean-Atmosphere General Circulation Model3,
Yukimoto et al. 2012) performs better in simulating diurnal rainfall over subtropical China (Yuan et al. 2013) and has the
finest resolution of $1.121° \times 1.125°$. Thus we select MRI-CGCM3 data applied in Poyang Lake Watershed to test the
performance of the STDDM.
The future data of MRI-CGCM3 includes simulations of the Representative Concentration Pathways (RCPs) of 8.5,6, 4.5 and
2.6. Compared to the other RCPs, in the RCP8.5 scenario temperature increases the most, which is corresponds to a highest
greenhouse gas emission, leading to a radiative forcing of 8.5 W/m$^2$ and temperature increase of 7.14 °C at the end of 21st
century (Taylor et al. 2012). The research is to detect the remarkable precipitation changes under climate warming; thus we
selected future simulations in the RCP8.5 scenario. In the study, we merge the historical (from 1961 to 2005), historical extent
(from 2006 to 2012) and RCP85 (from 2013 to 2100) data, as the merged data (1961-2100). To quantitatively analyze the
precipitation changes under climate warming in the 21st century, we compared precipitation between the baseline and future
period. As annual precipitation observations have main oscillation periods of quasi-20 years (Zhan et al. 2011), we selected
three 20 years from the merged data. From the merged data, simulations from 1998 to 2017 were selected as the baseline period
data, simulations from 2041 to 2060 were selected as the near future period data, and simulations from 2081 to 2100 were
selected as the further future period data.
The local grid observations (Hutchinson et al., 1998a; Hutchinson et al., 1998b; Zhao et al., 2014) with a resolution of $0.5° \times 0.5°$
are downloaded from the China Meteorological Data Service Center (http://data.cma.cn/). The local grid observations and
MRI-CGCM3 historical simulations were used to construct a relationship to correct the GCM data. China metrology point data
were also downscaled and used to validate the grid observations and the downscaled GCM simulations. To investigate the
relationship between precipitation changes and the temperature increment, we extract not only precipitations but also
temperature.
**3 Methodology**
**3.1 Future climate projection based on the spatiotemporally distributed downscaling model**
Considering the spatiotemporal heterogeneity of precipitation at the regional scale such as the Poyang Lake Watershed, we
developed a spatiotemporally distributed downscaling model (STDDM), which is a logical framework based on a specific
mathematic algorithm. The mathematic algorithm was used to create a mapping relationship between the global-scale GCM
simulations and the local scale climates variables. The mapping relationship is used as a transform function to correct the
future climate simulations to no-bias data. In the framework, we constructed respective mapping relationships between the
match-ups of GCMs simulations and local climate observations in each time (e.g., months or seasons) at each location. The
STDDM was improved compared to the traditional downscaling methods by adjusting the specific downscaling algorithm to
be suitable in the distributed space and time. Therefore, the downscaling processes show spatiotemporal differences in the
parameters or the equations, and the output data are spatially continuous, unlike that in traditional downscaling methods, which
ignores the temporal and continuous spatial differences and express space as discrete points instead of continuous grids.
Figure 2a shows the logical framework of the STDDM while Fig. 2b demonstrates how it was applied in Poyang Lake
Watershed using MRI-CGCM3 based on a linear-scaling algorithm. The STDDM contains three parts (Fig. 2a and b): (1)

upsampling GCMs simulations and local-scale observations to a continuous grid space of the same finer resolution; (2) constructing respective mapping relationship between the GCMs simulations and local observations in distributed space and time; (3) correcting the GCMs simulations using the mapping relationship constructed in step 2.

### 3.1.1 Upsampling GCMs simulations

MRI-GCM3 simulations were interpolated by Natural Neighbor Interpolation (Sibson et al., 1981) to a scale of 20 km×20 km, the smallest size of the subbasin of the Poyang Lake Watershed (Zhang et al. 2017), generating 263 spatial grids (Fig. 2b). For the spatiotemporally distributed downscaling, we used China meteorology spatially continua grid data as observations, instead of China meteorology station data. We interpolated the gridded observations to 20 km × 20 km, the same as the downscaled climate simulations. The match-up grids of simulations and observations at each time and each grid-box were generated.

### 3.1.2 Constructing relationships between the GCMs simulations and local observations

Because there is an inevitable mismatch between the simulations and observations (Li, 2009; Wood et al., 2004) after the upsampling, bias correction should be performed. The bias correction was processed by the transform function between match-ups of the upsampled simulation and observations, which represents the mapping relationship between the match-ups. The transform function could be any bias corrected model, including linear scaling, local intensity scaling, power transformation, distribution mapping models (Teutschbein et al. 2012) and other linear or nonlinear regression models.

As the influencing factors on climates show heterogeneity in space and time, we created spatiotemporally distributed relationships, described by the following formula.

$$C'_{T,S} = F_{T,S}(C_{T,S}) \qquad (1)$$

Where, $C'_{T,S}$ and $C_{T,S}$ indicate the upsampled global-scale climate simulations and the local climate variables, respectively, in the given time of $T$ and the space of $S$. $F_{T,S}$ demonstrates a transform function, used to correct the upsampled GCMs simulations. The function is a specific bias correction model, spatiotemporally distributed in mathematic equations or parameters, which is constructed based on the data from the historical period of 1961 to 2005.

In this study, we use a linear-scaling algorithm (Lenderink et al., 2007) as the bias correction model. For the linear-scaling
algorithm, the simulations were corrected by the discrepancy between the simulations and observations. Precipitations derived
from the GCMs were corrected by multiplying the precipitation bias coefficient, which is the ratio of the mean monthly
observation to simulation from the historical period; temperatures were corrected by adding the temperature bias coefficient,
which is the difference between the mean monthly observation and simulation in the historical period. However, as the bias
varies among the months from January to December and among the locations of the 236 spatial grids, a global standard bias
coefficient is prohibited. To better capture the bias in distributed time and space, we should create an individual bias coefficient
for the given month and gird box. Thus, a spatiotemporally distributed bias matrix was constructed. The respective downscaling
model and bias coefficient for a given month ($T$) and space ($S$) were established by Eq. 2 and 3.

$$P' = P \times P\_Cof \tag{2}$$

$$TM' = TM + TM\_Cof \tag{3}$$

where $P$ ($T$) represents the precipitation (or temperature) of upsampled simulations. $P'$ ($TM'$) represents the downscaled result
or upsampled observations; $P\_Cof$ ($TM\_Cof$) represents the bias correction coefficient of precipitations (or temperatures). In
the construction of $P\_Cof$ ($TM\_Cof$), $P$ ($TM$) and $P'$ ($TM'$) was set as the average monthly precipitation (or temperature) over
the historical time from 1961 to 2005. All the input and output data in the equations are in the given month ($T$) and space ($S$).
**3.1.3 Correcting the GCMs simulations**
The constructed relationship between the GCMs simulations and the observations from the historical period (in section 3.1.2)
also hold for the future (Maraun et al., 2010). Thus, the transform function was used to correct the future CGCMs simulations.
In this study, we corrected the daily and monthly precipitations (or temperatures) from MRI-CGCM3 by adding (or multiplying)
the bias coefficients in the corresponding month and grid box.

### 3.2 Precipitation changes analysis

### 3.2.1 Statistic indexes of precipitation changes

To obtain the general change in the temporal distribution, we calculated monthly precipitations from 1998 to 2100, averaged over the whole watershed. As floods and droughts occur more frequently in wet and dry months, we specifically analyze the extreme wet and dry precipitation changes in the 21st century. Therein, monthly precipitations, > 75% percentile of the 12 months, were classified as the extreme wet monthly precipitations for each year of the 103 years; monthly precipitations, ≤ 25% percentile were classified as the extreme dry monthly precipitation. The monthly precipitation of the 25%-50% and 50%-75% quantiles were classified as normal dry and wet monthly precipitations. The wet monthly precipitations include extreme and normal wet monthly precipitations; the dry monthly precipitations include extreme and normal dry monthly precipitations. To understand precipitation dynamics in terms of frequency and intensity, daily precipitations were categorized into five classes based on the classification by the Chinese Meteorological Administration and the possible risk of floods and droughts: light rain, medium rain, heavy rain, rainstorm, and extreme rainstorm with daily precipitation of 0.1-10, 10-25, 25-50, 50-100 and >100 mm/day, respectively. The frequency of precipitation intensities indicates heterogeneity in temporal distribution. The higher frequency of moderate rain means the more homogeneous, vice versa is the extreme rain. Therefore, the precipitation intensities were separated to moderate or extreme rains, including light rain, median rain or heavy rain, rainstorm, extreme rainstorm, respectively.

To further analyze the changes in precipitation frequencies and intensities, we calculate the annual days of light rain, medium rain, heavy rain, rainstorm and extreme rainstorm from 1998 to 2100 averaged over the whole watershed. Annual total precipitation, annual dry days, annual max daily precipitation and annual max continuous dry days were displayed as well. The meteorological stations (Fig. 1a) are uniformly distributed in the whole watershed and cover all kinds of the topographies and land covers. Therefore, in the study, the all above precipitation indexes of one year for the whole watershed were calculated based on the precipitation averaged over the grids containing the 15 stations, instead of the entire grids. Under global climate warming, precipitation becomes more concentrated which leads to more heterogeneity in temporal and spatial distribution (Donat et al., 2016; Min et al., 2011). Thus, we calculated variation coefficients for each year from 1998 to 2100, to investigate

the precipitation changes in temporal and spatial distribution. The variation coefficient measures the standard dispersion of the
data items, which can indicate the unevenness of temporal and spatial distributions of the precipitation. In this study,
heterogeneity in temporal, spatial and spatiotemporal distributions was measured by the temporal, spatial and spatiotemporal
variation coefficient, respectively. Temporal variation coefficients were calculated on the daily or monthly precipitations in
one year and the variation coefficient for one year is averaged over those of the 15 stations. For monthly precipitation, we only
select extreme wet and dry precipitations, as the extreme wet and dry are more likely to cause floods or droughts and thus
should be paid more attention. Spatial variation coefficients were calculated on the annual total precipitations of the 15 stations
in one year. Spatiotemporal variation coefficient was calculated on the monthly precipitations of the extreme wet months of
the wet stations and the extreme dry months of the dry stations in one year, as the extreme precipitation values were more
likely to cause floods or droughts.
**3.2.2 Relationship analysis between precipitation changes and temperature increasing**
We investigated the precipitation changes as a result of global temperature increase. To this end, we made liner regression
between the precipitation index and temperature changes from 2005 to 2100. We note that a mean filter with a window size of
21 years can reduce potential random fluctuation from precipitation by the most; thus was used to smooth annual precipitation
indexes and temperature simulations from 2005 to 2100. The long-time smoothed annual precipitation or temperature minus
the average annual value from 1998 to 2017, are set as precipitation index or temperature changes. A linear regression model
was used to investigate whether precipitation changes are related to climate warming. The two 11 years, 2005 to 2015 and
2090 to 2100 at the start and end, did not have filter diameter of 21 years; thus climate data used to be regressed is from 2016
to 2089.
**4 Result and Discussion**
**4.1 Model assessment**
Validation about the China meteorological grid observations should be performed, as well as the STDDM. As the STDDM
introduce the China meteorological grid observations and the grid data is not the direct in-suit data, validation about the gridded
data is necessary. The determination coefficient (R2), root mean square error (RMSE) and PBias (percent bias) were used to
examine the model performance.

### 4.1.1 Evaluation for the gridded meteorological

The China meteorological grid observations are referenced data to corrected GCMs simulations and reliability of the
observations is vital to the performance of the STDDM. So we make a validation using meteorological station observations,
in Fig. 3.
As shown in Fig. 3, we select four meteorological stations. The selected stations are uniformly distributed. The validation
produced an acceptable precision with $R^2 > 0.91$, absolute PBias < 2% for precipitations and $R^2 = 0.99$, absolute PBias < 6%
for temperature. All the dots of gridded and stationed value were distributed along the 1:1 line, thus confirming the satisfactory
performance.

### 4.1.2 Validations of precipitation and temperature projections in Poyang Lake Watershed

Before being used in future climate projection, the model should be examined. Data from 1961 to 1985 were used to construct
the model, and the remaining historical data from 1986 to 2005 were used to validate.
To test whether the downscaling method (STDDM) is effective in climate projections, we compare the results before and after
the bias correction in Fig. 4. The results before and after the bias correction marked as the outcomes by the STDDM and No-
STDDM, respectively. The projections by the STDDM show better performance with high correlations and narrow bias,
compared to the result by No-STDDM. Considering the complexity of climate physical mechanism and difficulty to accurately
simulate by the present methods, the uncertainty could be acceptable.
Using the STDDM and MRI-CGCMs, we obtained the temporal and spatial variation of future precipitations in the Poyang
Lake Watershed, and investigated the heterogeneity changes of precipitation in the temporal and spatial distribution.

### 4.2 Temporal variation of future precipitation

To discover the temporal variation under the future climate warming, we analyzed the monthly and daily precipitation changes
during the period from 1998 to 2100. For monthly precipitation, we analyzed intra-annual and inter-annual dynamics of
precipitation; based on the dynamics, we investigated the heterogeneity changes of monthly precipitation. For daily
precipitation, we analyzed the changes of precipitation intensities and frequencies; based on the changes, heterogeneity
changes of daily precipitation was also investigate.

**4.2.1 Monthly precipitation changes**

We analyzed the monthly precipitation changes during the period from 1998 to 2100 in Fig. 5. Precipitation show significant
intra-annual dynamics. Months with abundant rain (wet months), indicated by a reddish color, are mainly in April to July (the
wet season), while the rain-poor months (dry months), indicated by a bluish color, are mainly in September to the subsequent
February (the dry season). Precipitation concentrates in spring (March to May) and summer (July to August), occupying 73%
of the annual amount. The intra-annual dynamics of precipitation is similar to that shown by Feng (2012). Precipitation also
showed inter-annual dynamics. The wet months become wetter, and the wet season comes earlier from April to March, even
in February. In addition, each monthly precipitations of seven months (April to November) took increasing trends, of which
most months (5 months; April, May, June, August) are in the wet season; while precipitations of the other five months
experienced decreasing trends, all of which were in the dry season. It seems that wet months become wetter and dry months
become drier, in general.
To better demonstrate the inter-annual dynamics of precipitation, monthly precipitations in each year were sorted in a
descending order in Fig. 5(b). As the time of the monsoon reaching the Poyang Lake Watershed, varied in different years, with
1~2 months' advance or delay; the wet or dry months for different years are not the same. By sorting monthly precipitation,
wet months and dry month could be distinguished intuitively in Fig. 5(b). Obviously, monthly precipitation of wet months
experienced an increasing trend respectively, even some with slight significance; in contrast, each dry monthly precipitation
exhibited decreasing trends, separately, despite the insignificant signs. We accumulated the extreme wet or dry monthly
precipitations for each year in Fig. 6. The precipitation of extreme wet months showed a significantly increasing trend ($p<0.05$)
(Fig. 6a), while the precipitation of the extreme dry months demonstrated a significantly decreasing trend ($p<0.05$). Extreme
wet months increased from 277.82 mm•month$^{-1}$/a over historical time from 1998-2017, to 344.10 mm•month$^{-1}$/a over future
time from 2081 to 2100, by 23.86% with a change rate of 7.3 mm•month$^{-1}$/10a. Extreme dry months decreased from 35.44
mm•month$^{-1}$/a over historical time from 1998-2017, to 30.46 mm•month$^{-1}$/a over future time from 2081 to 2100, by -14.05%
with a change rate of 0.92 mm•month$^{-1}$/10a. Therein, the extreme wet months are mainly concentrated in March-July (Fig. 6c),
part of the wet season, while the extreme dry months are mainly concentrated in September-February (Fig. 6d), consistent to
the dry season.
Overall, under climate warming over the 21st century, the wet monthly precipitations become wetter while the dry month
precipitations become drier, which suggested the uneven temporal distribution of precipitation (Fig. 7). As shown in Fig. 7,
the temporal variation coefficient of the extreme month (including extreme wet and months) precipitations within each year
from 1988 to 2100, experiences significantly increasing trends (p<0.01), and increased from 0.76 /a over historical time from
1998-2017, to 0.84 /a over future time from 2081 to 2100, by 10.53% with change rate of 0.01 /10a. The significantly increasing
trends indicated the more uneven trend of precipitation in the temporal distribution, which might lead to increased  risks of
floods and droughts.
**4.2.2 Daily precipitation changes**
To understand the changes in precipitation intensities and frequencies under future climate warming, daily precipitation
variations were also analyzed and are shown in Fig. 8. Moderate vs extreme rain frequencies (Fig. 8a and b), the annual total
rain vs the annual total rainy days (Fig. 8c), and the annual max precipitation vs the annual max continuous rainy days (Fig.
8d) were analyzed.
Under climate warming, the annual frequency of moderate rains experienced decreasing trends; in contrast, the annual
frequency of extreme rains experienced significantly increasing trends (Fig. 8a). Statistically, averaged over 103 years, annual
precipitation frequencies are dominated by the moderate rain frequency a total of 163.70 days, or 44.8% (163.70/365), while
the extreme rain occurs less often, a total of 20.70 days, or 6.70% (20.7/365). The remaining is rain-free days, a total of 180.75
days, 49.5% (180.75/365). The annual moderate rain frequency decreased, from 170.56 days/a over the historical period from
1998 to 2017, to 159.55 days/a over  the future period from 2081 to 2100, by -6.46% with a change rate of -14.4 days/10a; on
the contrary, the annual extreme rain frequency increased from 19.18 days/a over historical time from1998 to 2017, to 23.42
days/a over future time from 2081 to 2100, by 22.10% with a change rate of 0.49 days/10a (Fig. 8b).
Furthermore, the annual total rainy days, the sum of the moderate and extreme rain frequencies, demonstrated a significantly
decreasing trend in the 21st century, whereas the annual total precipitation exhibited a significantly increasing trend (Fig, 7c).
Rainy days decreased from 187.57 days/a over the historical period from 1998 to 2017, to 180.37 days/a over the future period
from 2081 to 2100, by -3.84% with a change rate of -1.00 days/10a; while the annual total rain amount increased, from 1650
mm/a over the historical period, from 1998 to 2017, to 1906 mm/a over the future period, from 2081 to 2100, by 15.55% with
a change rate of 23.00 mm/10a. The increase in the annual total rain and decrease in the annual rainy days suggested more
concentrated precipitation and dry days in the future. This tendency might lead to the increased risk of floods and droughts,
which was also indicated by the increased annual max daily precipitation and max continuous dry days (Fig. 8d). Annual max
daily precipitation increased from 148.76 mm•day$^{-1}$/a averaged over the historical period from 1998 to 2017, to 212.01
mm•day$^{-1}$/a averaged over the future period from 2081 to 2100, by 42.51% with a change rate of 7.2 mm•day$^{-1}$/10a; while the
max continuous dry days increased from 25.35 days/a over the historical period from 1998 to 2017, to 28.15 days/a over the
future period from 2081 to 2100, by 11.05% with a change rate of 0.5 days/10a.
Overall, the significantly inverse change trends in the moderate vs extreme rain frequencies, the annual total rain vs the annual
total rainy days, and the annual max precipitation vs the annual max continuous rainy days, indicated an increasing temporal
heterogeneity in precipitation distribution over the 21st century. Obviously, the increasing heterogeneity was exhibited by the
increasing temporal variation coefficient of daily precipitations (Fig. 9). The temporal variation coefficient of daily
precipitations increased from 1.50 /a over the historical period from 1998 to 2017, to 1.62 /a over the future period from 2081
to 2100, by 7.48% with a change rate of 0.016 /10a.
**4.3 Spatial variation of future precipitation**
Climate warming could cause the rain belt shift (Putnam et al., 2017), which might lead to precipitation changes in the spatial
pattern. To investigate the spatial variation, we analyzed the similarities and differences of precipitation changes in space (Fig.
1); based on the differences, we use the indexes of the spatial and spatiotemporal variation coefficient to investigate the spatial
heterogeneity changes (Fig. 11). Fig. 10 shows the precipitation changes in the spatial pattern during the period from 1998 to
2100; Fig. 11 shows the spatial and spatiotemporal variation coefficient for each year over 1988 to 2100.
Precipitations showed a regular spatial pattern both in the wet and dry season, in Fig. 10a-c and e-g. More specifically,
precipitation was distributed more in the east and west, however less in the north central plain and the south bottom depression.
Rich rain in the east and west are dominated by the southeast and southwest summer monsoons. Less precipitation was due to
the leeward sloop of the eastern (Xuefeng Mountain) and western mountains (Wuyi Mountain). Less precipitation in the south
bottom depression was because that water vapor was blocked from this region by the NanLing Mountain in the south (Fig. 1a).
The precipitation distribution in spatial pattern from 1998 to 2100 (Fig. 10 a-c and d-f) were consistent with the observations
from 1951 to 2005 (Fig. 1b.), thus confirming the satisfactory performance of the STDDM. Moreover, wet and dry season
precipitation showed inverse changes. The wet season precipitations exhibited ascending (Fig. 10a-c and g) change while the
dry season precipitation exhibited descending (Fig. 10d-f and h) change from 1998 to 2100. The inverse changes were
consistent with the interannual variability of increased precipitation in wet months and decreased precipitation in dry months
(Section 4.2). The increase of precipitation in the wet seasons and decrease in precipitation in the dry seasons were also detected
in the change rate of the cells over the entire watershed (Fig. 10g or h).
However, precipitation change also showed a different spatial pattern. Precipitation change rate was heterogeneous in spatial
distribution for dry or wet season respectively (Fig. 10g and h). In the wet season, the precipitation increased more in the north
part of the watershed, except for the central plain (Fig. 10g); in the dry season, the precipitation decreased more in the central
area (Fig. 10h). Statistically, in the wet season, precipitation increased with the change rate raising from $\leq 3.6$ mm/10a in the
southwest, to $\geq 11.7$ mm/10a in the northeast; in the dry season, precipitation decreased with the change rate falling from $\geq$ -
2.0 mm/10a in the surrounding region, to $\leq$ -2.7 mm/10a in the central region. Furthermore, precipitation changes show
different spatial characteristics in wet and dry seasons. From 1998 to 2100, in the wet season (Fig. 10a-c), the wet area (the
reddish area, mainly in the north except for the center plain) becomes wetter; in the dry season (Fig. 10 d-f), the dry area (the
bluish area, mainly in the north center plain and in the south depression) become drier.
The uneven change rates may lead to increase of the spatial heterogeneity of precipitation under global warming, and the
tendency of the wet area to become wetter and the dry area to become drier also indicated the increasing spatiotemporal
heterogeneity of precipitations. Indeed, the spatial heterogeneity did increase, with the spatial variation coefficients raising
from 0.097 /a over the historical period (1998-2017), to 0.110 /a over the future period (2081−2100), by 12.64% with a change
rate of 0.002 /10a (Fig. 11a). The spatiotemporal heterogeneity did increase with the spatiotemporal variation coefficient
raising from 0.89 /a over the historical period (1998-2017), to 0.94 /a over the future period (2081-2100), by 4.96% with a
change rate of 0.008 /10a. Overall, the uneven change rates for the whole basin and inverse changes for the dry and wet area
indicated an increasing spatial heterogeneity in precipitation distribution over the 21st century.
**4.4 The impact assessment of temperature increment on precipitation changes**
Previous studies have detected precipitation changes and have attributed these changes to climate warming (Westra et al., 2013;
Zhang et al., 2013). In this study, the spatiotemporal changes of precipitation in the Poyang Lake Watershed in the 21st century
were hypothesized to be related to temperature increments. So we analyze the correlations qualitatively and quantitatively.
The following are trying to demonstrate the driving force related to climate warming on precipitation changes in the temporal
pattern. In the wet season from April to July, the summer monsoon might become weaker in Southeast Asia as the temperature
increasing (Wang, 2001; Wang, 2002; Guo et al., 2003). Consequently, the summer monsoon is delayed for a longer time in
the middle and lower Yangtze River basin instead of moving further north. The delay leads to much more rain during the wet
season. As being located in the middle of the Yangtze River basin, the Poyang Lake Watershed becomes wetter in the wet
season (Fig. 5-5, Fig. 10a-c). In fact, the increase in precipitation in the Poyang Lake Watershed was detected in previous
studies (Yu and Zhou, 2007; Ding et al., 2008). In the dry period from September to the subsequent February (especially in
the winter season, from December to February), during which summer monsoon is inactive, there is less water vapor in the
atmosphere to condense into rain. Additionally, stronger winds in the winter (Wu et al., 2013) blow the evaporation away, thus
enhancing the difficulty of generating rain from water vapor compared to the other seasons. When temperature increases, the
ability of the atmosphere to hold water vapors is strengthened, which makes it more difficult to precipitate. Therefore,
precipitation decreases in the dry season, consistent with Li et al.'s (2016) result. As temperature increment increases the ability
of the atmosphere to contain water vapor, rain is more difficult, and if it rains it will rain largely (Min et al., 2011; Zhang et
al., 2013). Thus, the frequency of heavy rain and rain-free events increases, indicating increased frequency and strengthened
intensity of the extreme precipitation. Overall, the climate warming might make precipitation more temporally uneven.
Climate warming could also explain the spatial distribution of precipitation changes in the dry and wet seasons. In the wet
season, the summer monsoon delays in the middle and lower Yangtze River Basin. The delaying area covers only the north
part of the Poyang Lake Watershed. As it receives abundant water vapor from the delayed summer monsoon, the north part of
Poyang Lake Watershed experiences a greater increase in precipitation with a larger change rate (Fig. 10g). The eastern Poyang
Lake Watershed is the nearest to the western Pacific Ocean; thus the eastern region receives more continuous water vapor. So
the precipitation change rate decreases from the southeast to the northwest in the wet season. However, in the dry season
especially in winter, during which there is a low-frequency or absent summer monsoon, the water vapor mainly comes from
evapotranspiration. In the watershed, the periphery is covered by the lake of Poyang in the northern plain and high-density
vegetation in the northwest, southeast and southwest mountains; so there is more evapotranspiration in the periphery. The
center is mainly covered by farmland and grassland; so there is less evapotranspiration in the center (Wu et al., 2013). Thus,
the moisture decreases from the surrounding to the center. Therefore, as temperature increases, it is more difficult for rain to
occur in the area of lower moisture, the center of the Poyang Lake Watershed. Therefore the precipitation decreased with a
change rate falling from the surrounding to the center in the dry season (Fig. 10h).
To quantitatively analyze the relationship between precipitation changes and temperature increment, we created a scatter plot
between precipitation indexes changes and temperature increment, as shown in Fig. 12. A trend analysis was conducted using
linear regression of each annual precipitation index over the 103 years from 1998 to 2100. The associated slopes represent the
change rate of each precipitation index relative to temperature increment. The significance of the trend is indicated by p value.
As shown in Fig. 12, there is a significant correlation between the precipitation change and the temperature increment, with p
$\leq 0.001$ and $R \geq 0.78$ for 6 precipitation indexes: the annual precipitation in the wet season (Fig. 12a), the annual max daily
precipitation (Fig. 12d), the temporal variation coefficient of the monthly precipitation (Fig. 12c), the temporal variation
coefficient of the daily precipitation (Fig. 12f), the spatial variation coefficient (Fig. 12g) and the spatiotemporal variation
coefficient (Fig. 12h). However, changes of the other two precipitation indexes, the annual precipitation in the dry season (Fig.
12b) and the annual max continuous dry days (Fig. 12e), appeared to be correlated with slight signs of $p \leq 0.05$ and $R \leq 0.58$.
The overestimation of moderate- or free-rain frequency  from the GCM simulations (Teutschbein et al. 2012) might explain
the slightly low correlation between the annual precipitation values in the dry season and temperature increment, while the
overestimation of the precipitation frequencies (Prudhomme et al. 2003) could explain the slightly low correlation between the
annual max continuous dry days and  temperature increment. For all the correlations (Fig. 12a-h), the precipitation changed
with fluctuation, which might be caused by random variations from GCMs.
Overall, despite the low correlations and stochastic fluctuation, the correlations could indicate that the climate warming can
partly explain the precipitation changes. Statistically, precipitation changes relative to temperature increment are 16.657
mm•month$^{-1}$/K, -4.31 mm•month$^{-1}$ /K, 17.45 mm•day$^{-1}$ /K, 0.71 days/K, 0.028/K, 0.033/K, 0.0074/K and 0.02/K for the annual
precipitation in the wet season, the annual precipitation in the dry season, the annual max daily precipitation, the annual max
continuous dry days, the temporal variation coefficient of the monthly precipitation, the temporal variation coefficient of the
daily precipitation, and the spatial variation coefficient and the spatiotemporal variation coefficient, respectively.
In summary, the explanation of precipitation changes in temporal and spatial distribution qualitatively and quantitatively,
suggests the downscaling method is reasonable and the STDDM could be applied in the basin-scale region based on a GCM
successfully.
**5 Conclusion**
A spatiotemporally distributed downscaling method (STDDM) was proposed in this study. The downscaling method
considered the heterogeneity in spatial and temporal distributions, and produced local climate variables as spatially continuous
data instead of independent and discrete points. The STDDM showed a better performance than the No-STDDM. Using the
STDDM, we constructed the spatially continuous future precipitation distribution and dynamics in the wet and dry season from
1998 to 2100, based on MRI-CGCM3. Several findings were obtained:
First, the spatial and temporal heterogeneity of precipitation increased under future climate warming. In the temporal pattern,
the wet season become wetter, while the dry season become drier. The frequency of extreme precipitation increased, while
that of the moderate precipitation decreased. Total precipitation increased, while rain days decreased. The max dry day number
and the max daily precipitation both increased. These precipitation changes demonstrated an increasing heterogeneity of
precipitation in temporal distribution, and the change rate of temporal heterogeneity is 0.01 /10a (0.016 /10a) for the temporal
variation coefficient of the monthly (daily) precipitation. In the spatial pattern, the change rate of precipitation was uneven
over the whole watershed. Additionally, the wet areas become wetter in the wet season and the dry areas become drier in the
dry season. The uneven change rates for the whole basin and inverse change for dry and wet area demonstrated an increasing
heterogeneity in the spatial distribution, and the change rate of spatial heterogeneity was 0.002/10a.
Second, precipitation changes can be significantly explained by climate warming, with $p < 0.05$ and $R \geqslant 0.56$. The explanation
of precipitation changes in temporal and spatial distribution qualitatively and quantitatively, suggests the downscaling method
is reasonable and the STDDM could be applied in the basin-scale region based on a GCM successfully.
The results can be applied to a hydrological and hydrodynamic model, to study the future changes in water volumes, lake
levels and areas response to climate warming. The relationship between precipitation variations and temperature increment
could be helpful to the driving forces analysis of precipitation changes. The dry to be drier and wet to be wetter condition may
lead to increased risk of floods and droughts. In particular, in the region where floods and droughts do not usually occur,
additional adaptation measures could be taken to prevent loss from the future frequent hydrological disasters.
**Data availability**
All data can be accessed as described in Sect. 2.2. The data sets and model codes are provided in the supplements.
**Acknowledgements**
This work was funded by the National Key Research and Development Program (2017YFB0504103), the National Natural
Science Funding of China (NSFC) (41331174), the Open Foundation of Jiangxi Engineering Research Center of Water
Engineering Safety and Resources Efficient Utilization (OF201601), the ESA-MOST Cooperation DRAGON 4 Project
(EOWAQYWET), the Fundamental Research Funds for the Central Universities (2042018kf0220) and the LIESMARS
Special Research Funding.

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

**Figures**

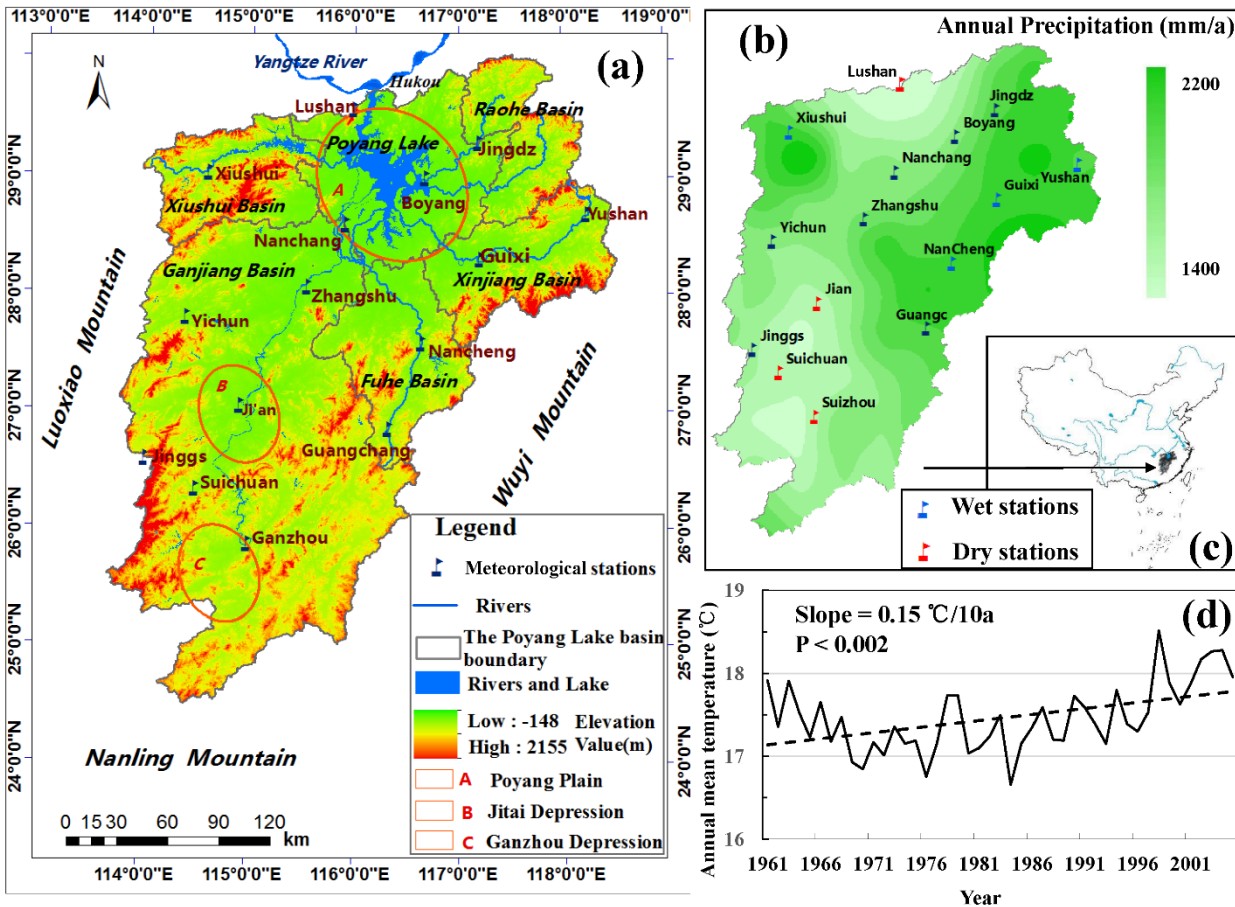


Fig. 1. The topography and landforms (a), precipitation distribution and dry-wet stations (b), temperature change (d) and
location of the Poyang Lake Basin (c). We sorted the annually accumulated precipitation of the 15 stations, averaged over time
from 1961 to 2005. The 4 stations with the max or min mean annual precipitations are set as dry or wet stations, respectively.

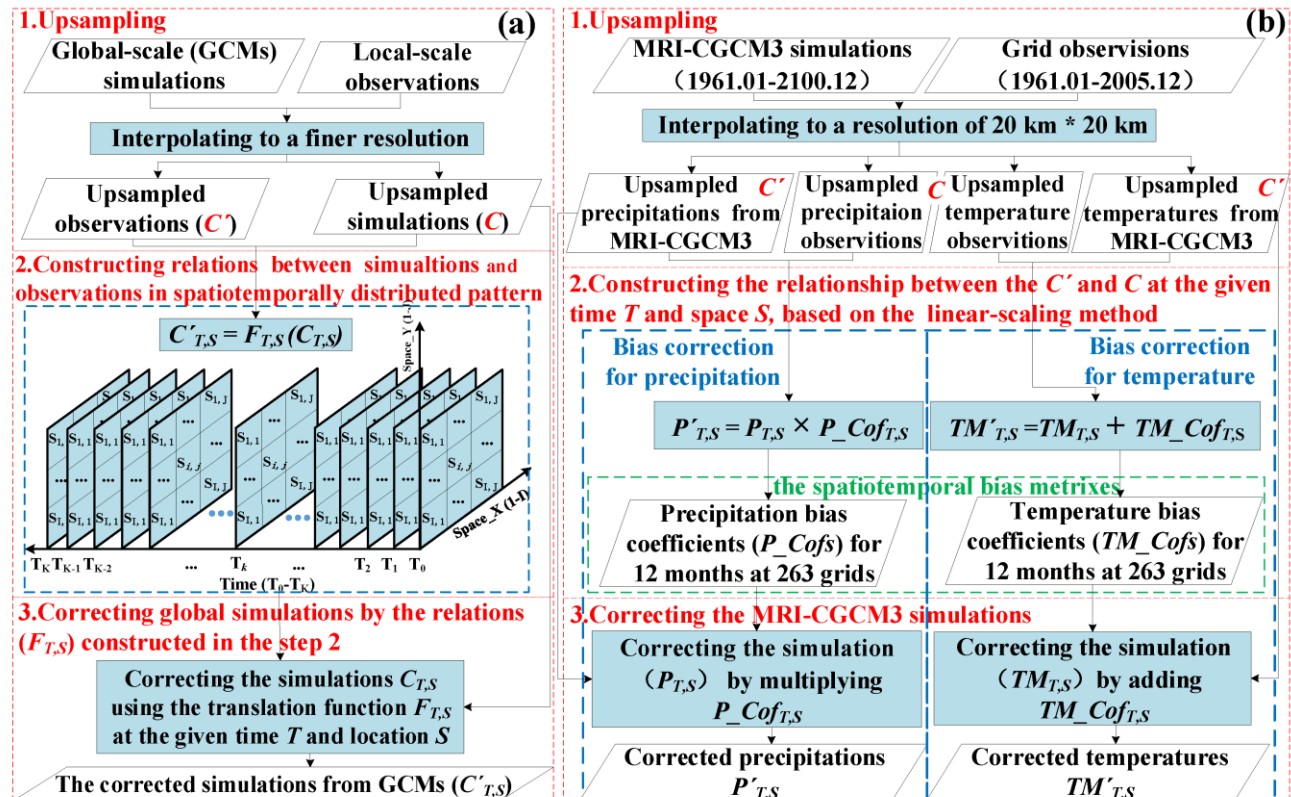


Fig. 2 Conceptual flow chart of the climate projection including upsampling, relation construction and correction: The
common framework of the STDDM (a) and test case base on the linear-scaling algorithm (b). The STDDM was used to project
MRI-CGCM3 simulations from 1998 to 2100.


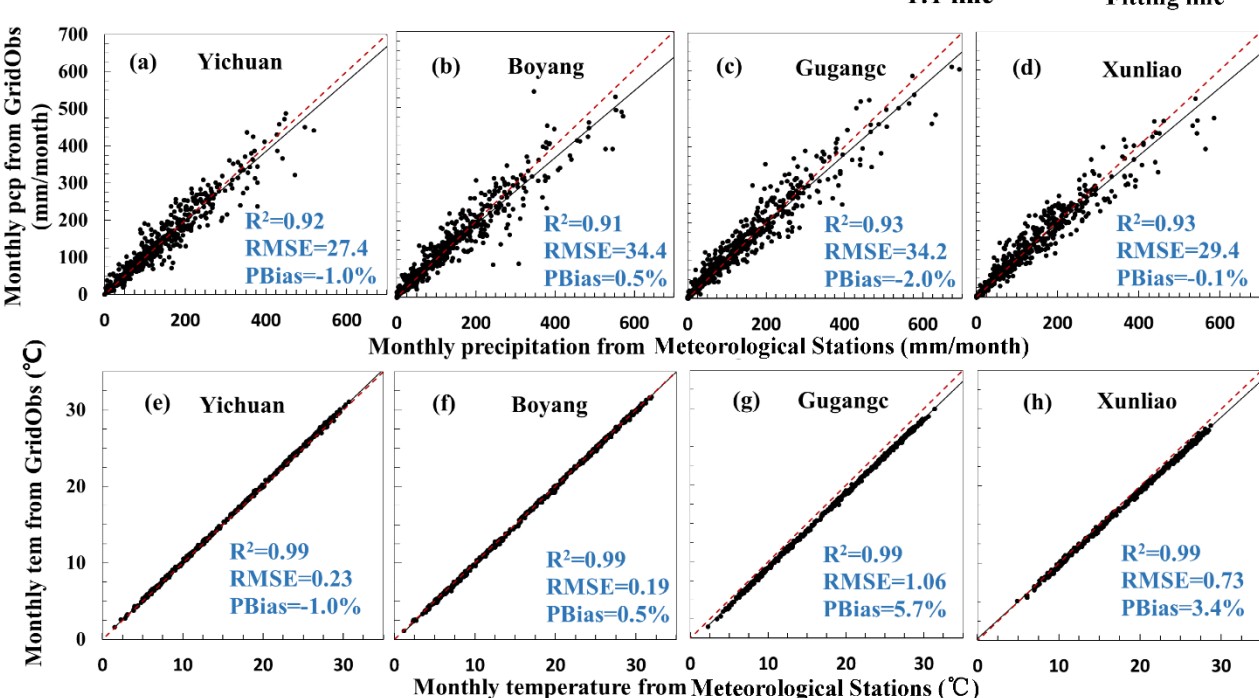


Fig. 3. Validation of gridded meteorological data (GridObs) by using gauging stations observation: Precipitation (pcp; a,b,c
and d) and temperature (tem; e,d,f and g) at meteorological station of Jian (a and e), Ganzhou (b and d), Zhangshu (c and f)
and Lushan (d and g).

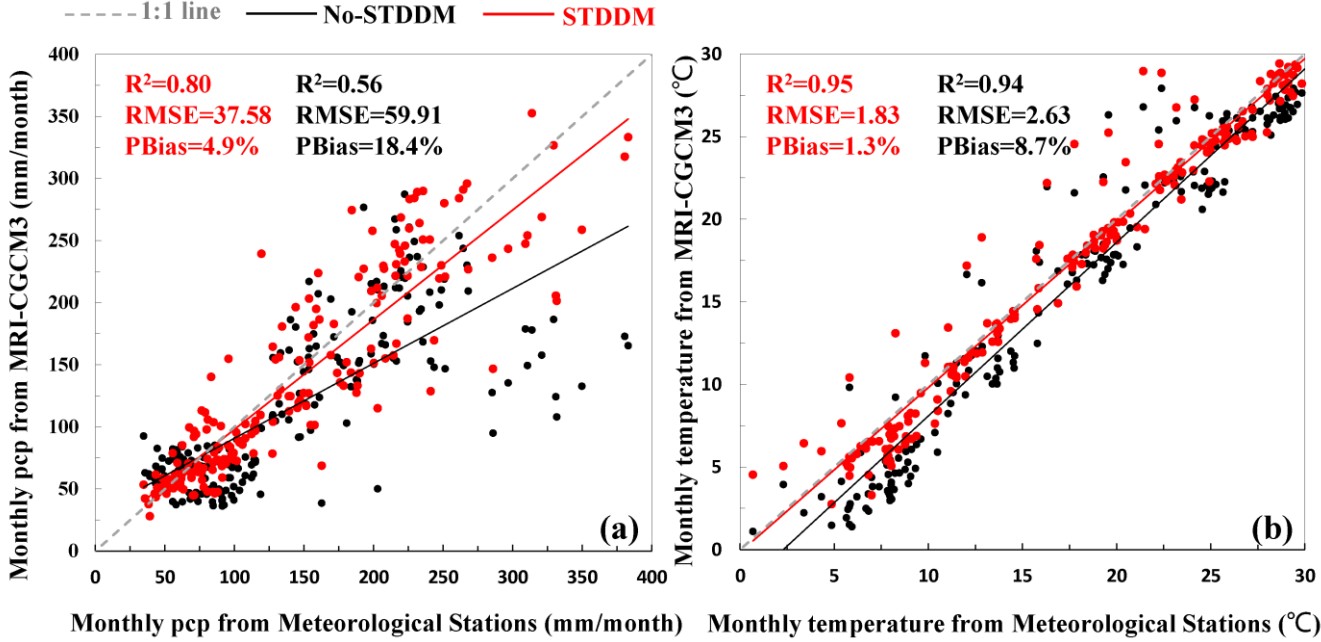


Fig. 4. Validation of the precipitation (pcp) (a) and temperature (b) projections by the STDDM (in black) and No-STDDM (in
red). Dots represent the monthly precipitations (or temperatures), averaged over 20 years from 1986 to 2005. The dots contain
monthly precipitations of the 15 stations. The solid lines represent linear regression which is the best fit through all match-ups
of the projections and observations.



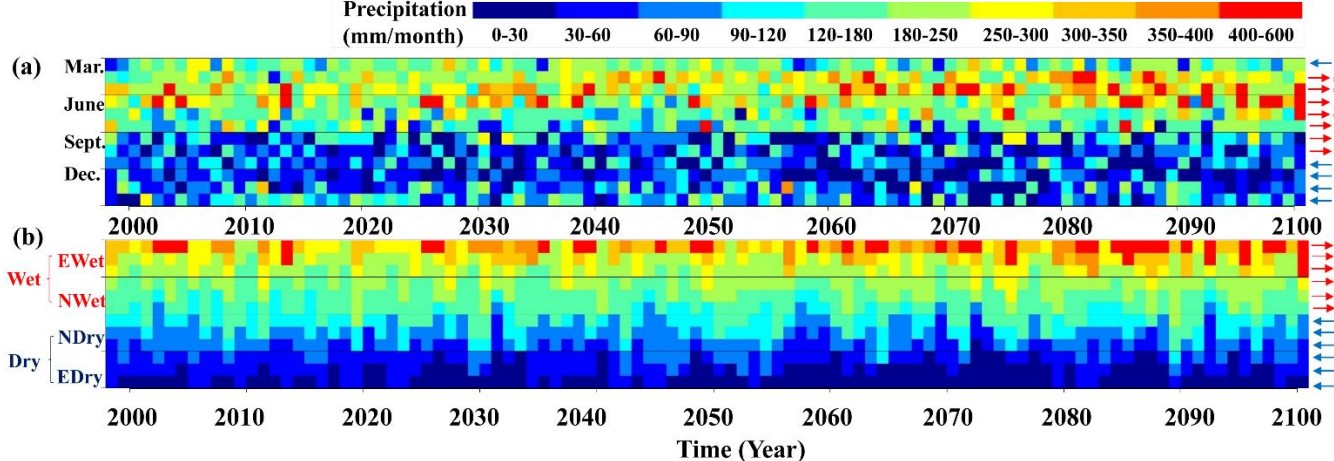


Fig. 5. Total variability of monthly precipitation from 1998 to 2100. Each column represents the data for one year and each
cell represents an accumulative precipitation of one month. The red (blue) arrows indicate that the monthly precipitation
experienced an increasing (decreasing) trend over the 103 years, respectively. The asterisk demonstrates the significant trends
with p<0.05. (a) Monthly precipitation in month order, referred to Spring (March to May), summer (June to August), autumn
(September to November), and winter (December to next February) from top to bottom, respectively. (b) Monthly precipitation,
sorted in the descending order for each year, where months are classified as extreme wet (EWet), normal wet (NWet), normal
dry (NDry) and extreme dry (Edry) months from up to down. Therein, wet months (Wet) include extreme and normal wet ones
while dry months (Dry) include extreme and normal dry ones.

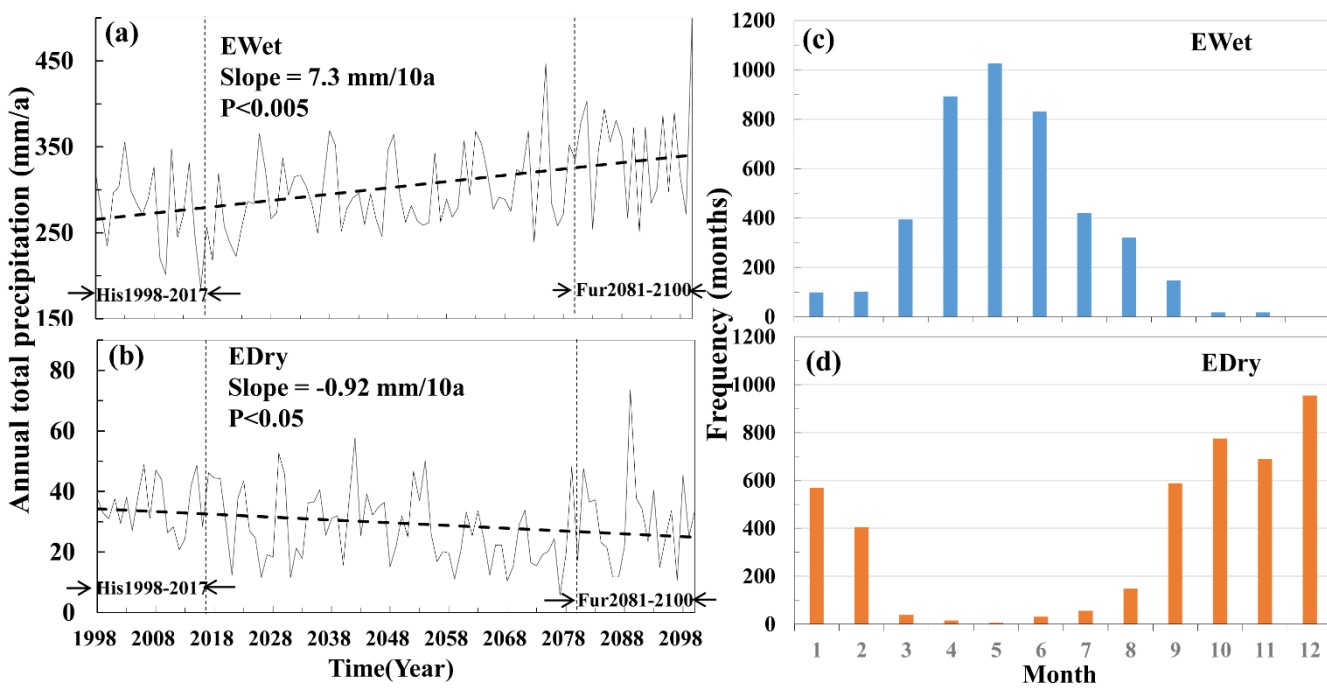


Fig. 6. The trends of changes in monthly precipitations of extreme wet (EWet) (a) and dry (EDry) (b) months from 1998 to
2100. The further future period from 2081 to 2100 (Fur2081-2100) and baseline period from 1998 to 2017 (His1998-2017) are
indicated by arrows. Frequencies of the months in extreme wet (c) or dry (d) months are calculated during the period from
1998 to 2100.

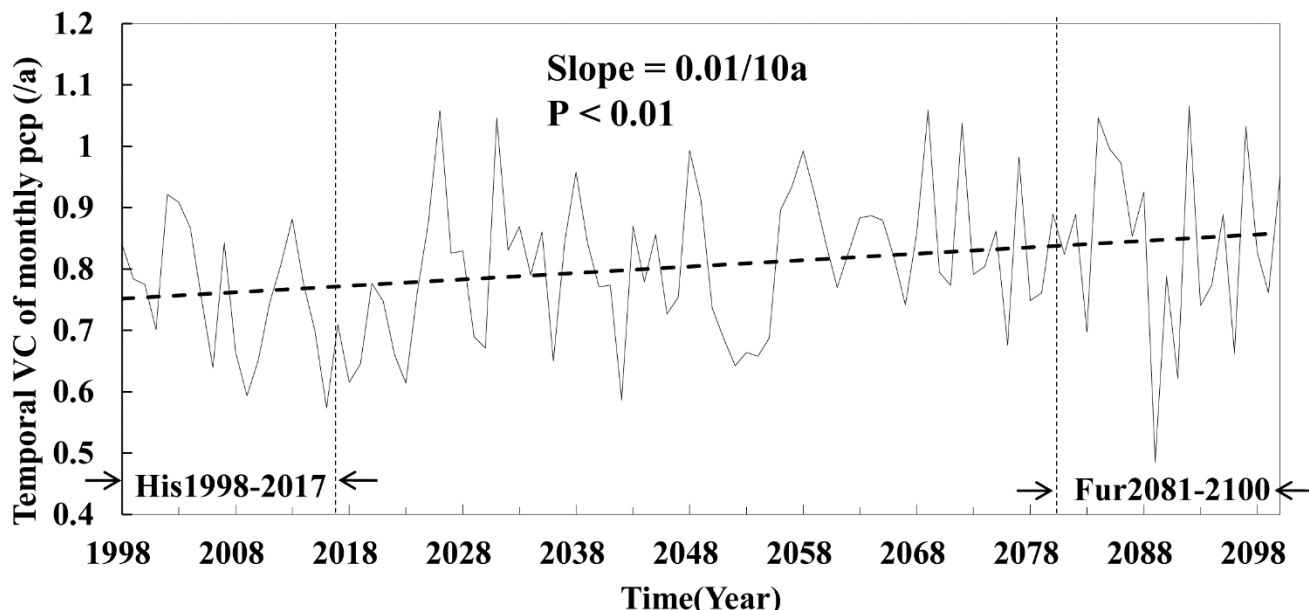


Fig. 7. The temporal variation coefficients of the extreme month precipitations for each year over 1988 to 2100. The extreme
months are composed of the extreme wet and dry months. The far future period from 2081 to 2100 (Fur2081-2100) and baseline
period from 1998 to 2017 (His1998-2017) are indicated by arrows.



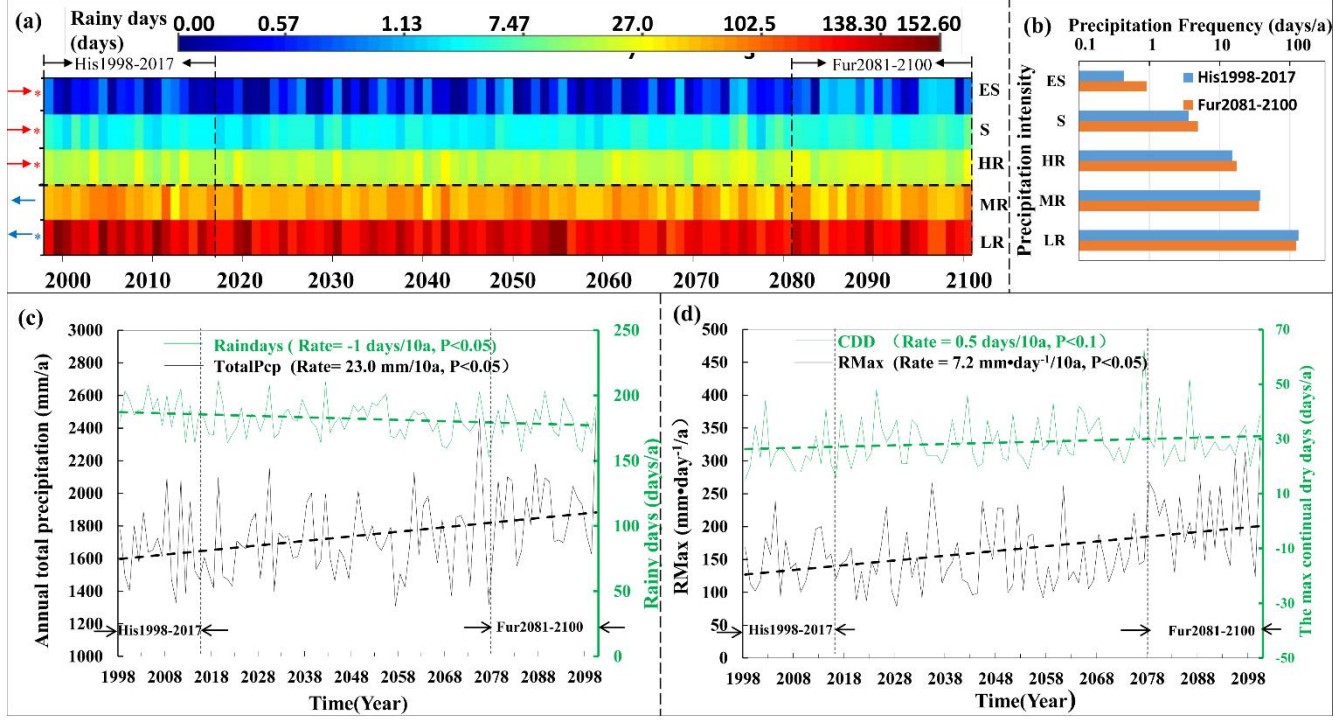


Fig. 8. The changes in daily precipitation intensities and frequencies. (a) Precipitation intensities and frequencies for each year
over 1998 to 2100, where each column represents a year and each row indicates a precipitation intensity. Daily precipitation
intensities are categorized to 5 classes, Light Rain (LR), Median Rain (MR), Heavy Rain (HR), Rainstorm (S), and Extreme
Rainstorm (ES) with daily precipitation of 0.1-10, 10-25, 25-50, 50-100 and >100 mm/day, respectively. The moderate rain
includes LR and MR while the extreme rain is composed of HR, S, and ES. The cell represents an annual frequency of one
precipitation intensity, with a unit of days. The red (blue) arrows indicate that annual frequency of the precipitation intensity
experienced an increasing (decreasing) trends over the 103 years (from 1998 to 2100), respectively. The asterisk represents
the significant trends with p<0.05. The far future period from 2081 to 2100 (Fur2081-2100) and baseline period from 1998 to
2017 (His1998-2017) are indicated by arrows. (b) Precipitation frequencies of LR, MR, HR, S, and ES for Fur2081-2100 and
His1998-2017, respectively. (c) The change of the long-term data for annual total precipitation (totalPcp) and total rainy days
(Raindays). (d) The change of the long-term data for annual max daily precipitation (RMax) and annual max continuous dry
days (CCD).

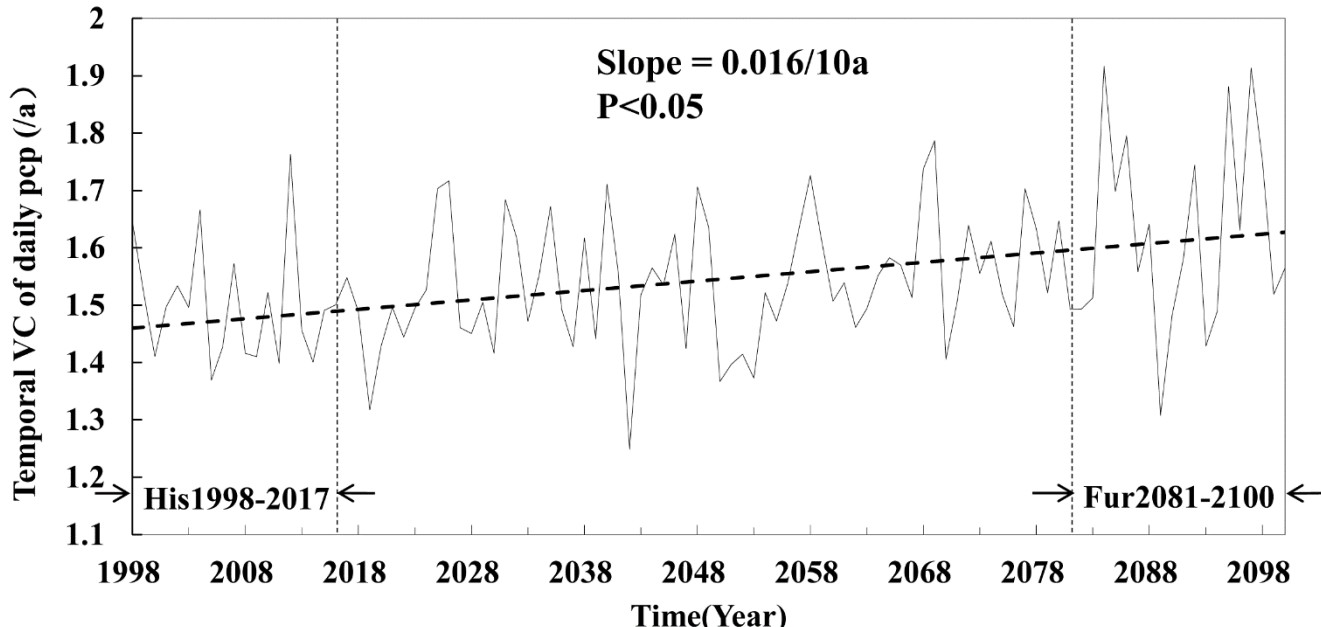


Fig. 9. The temporal variation coefficient of daily precipitations for each year over 1988 to 2100. The far future period from
2081 to 2100 (Fur2081-2100) and baseline period from 1998 to 2017 (His1998-2017) are indicated by arrows.

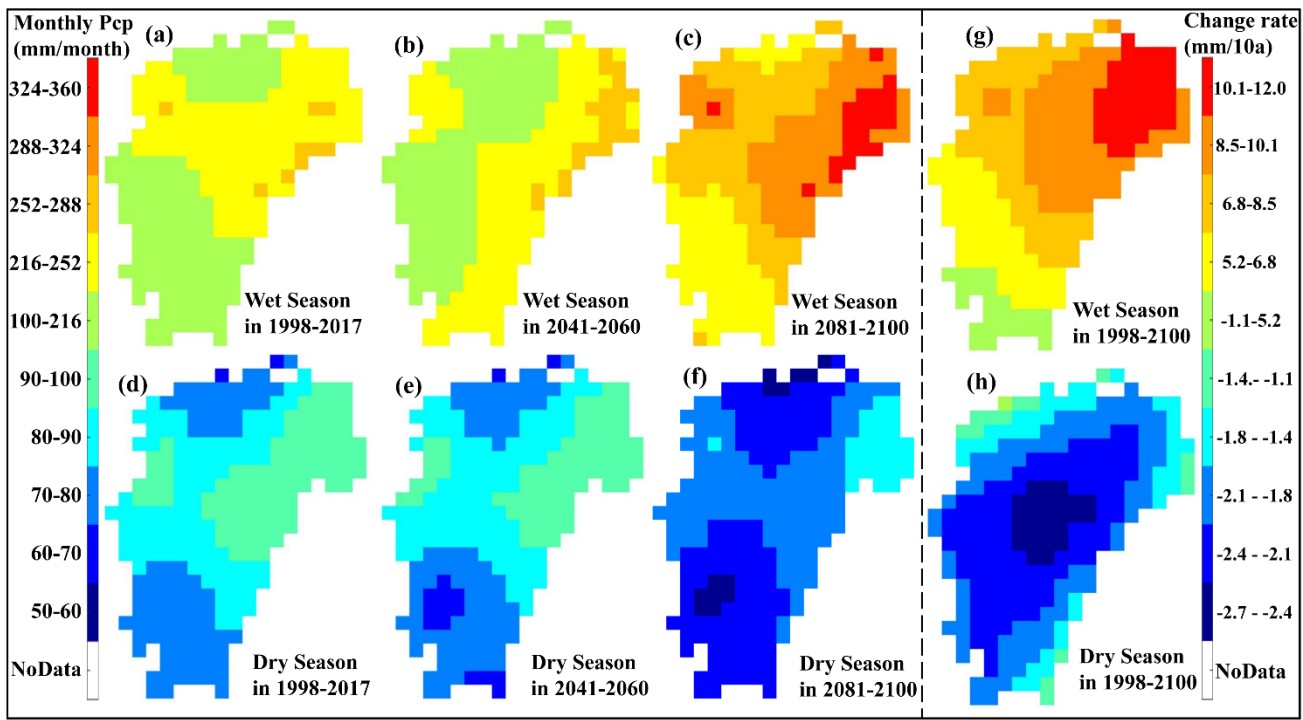


Fig. 10. The precipitation changes in the spatial pattern during the period from 1998 to 2100: average monthly precipitations
of the wet season (April to July) during the period from 1998 to 2017 (a), 2041 to 2060 (b), and 2081 to 2100 (c); average
monthly precipitations of the dry season (December to next February) during the historical period from 1998 to 2017 (d), 2041
to 2060 (e), and 2081 to 2100 (f); change rate of monthly precipitation in wet (g) and dry (h) season from 1998 to 2100. As
floods and droughts occur more frequently in extreme months, the precipitation in the analysis considered only the extreme
wet (April-July) and dry (September-February) months (Fig. 5c and d). Besides, precipitation is dominated by southeast
summer monsoon, which brings water vapor from the sea. The summer monsoon is frequent from the end of spring and stat
of autumn, covering the wet months April to July. However, though as dry months, the autumn period from September to
November is affected by southeast summer monsoon (Tan et al., 1994) slightly because autumns are the transpiration periods
of summer to winter. Therefore, winter (December-February) was represented as the dry season with poor rain; while April-
July was represented as the wet season with abundant rain.


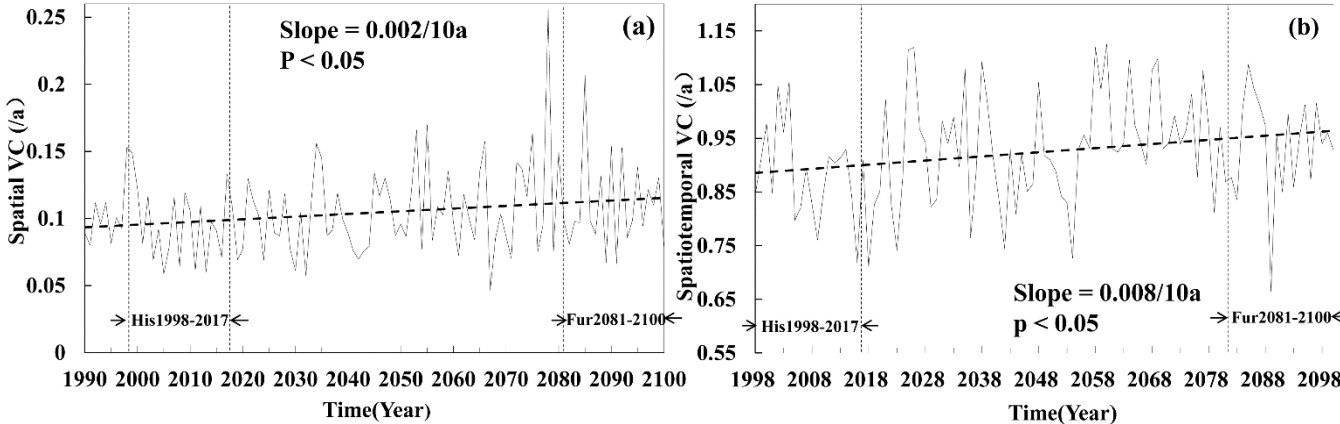


Fig. 11. The spatial (a) and spatiotemporal (b) variation coefficient for each year over 1988 to 2100. The further future period
from 2081 to 2100 (Fur2081-2100) and baseline period from 1998 to 2017 (His1998-2017) are indicated by arrows.





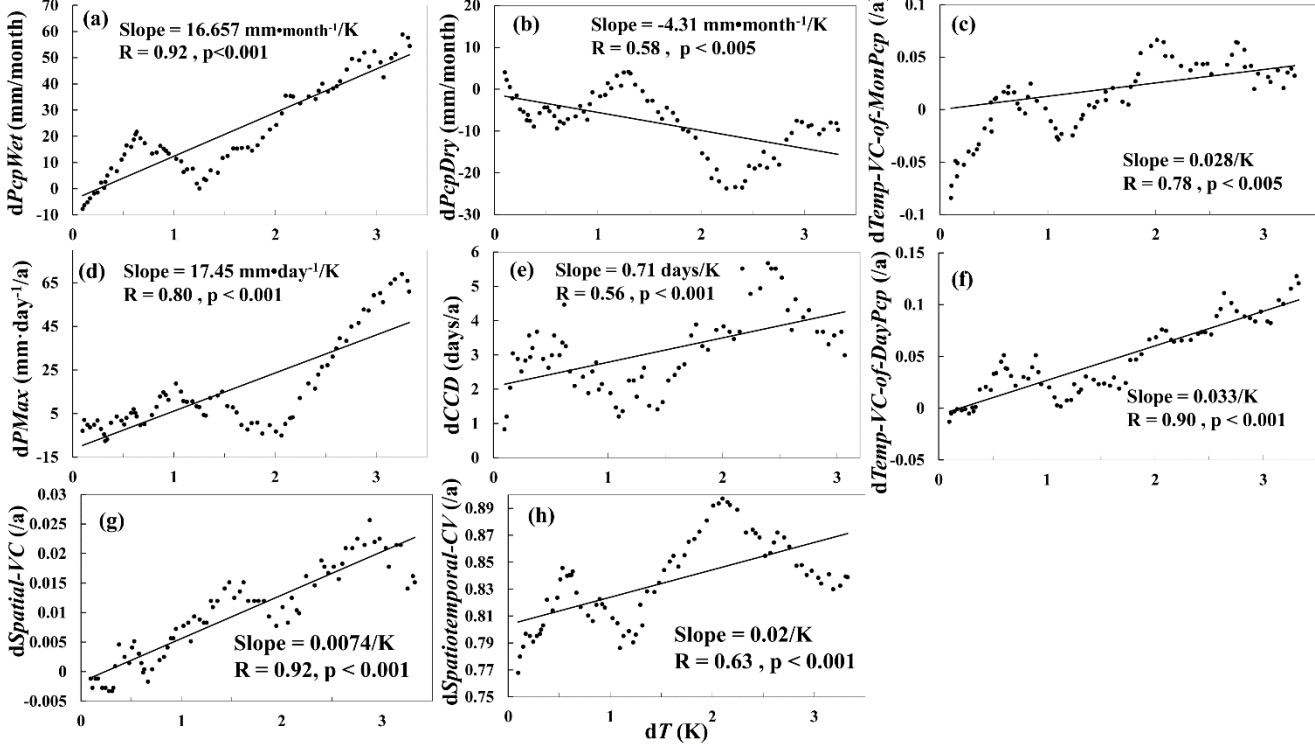


Fig. 12. The relationship between the precipitation index changes (d*PcpIndex*) and the temperature increment (d*T*). The precipitation indexes include annual precipitation in the wet season (PcpWet) (a), annual precipitation in the dry season (PcpDry) (b), temporal variance coefficient of monthly precipitations (Temp-VC-of-MonPcp) (c), annual max daily precipitation (PMax) (d), annual max continuous dry days (CCD) (e), temporal variance coefficient of daily precipitations (Temp-VC-of-DayPcp) (f), spatial variance coefficient (Spatial-VC) (g), and spatiotemporal variance coefficient (Spatiotemporal-VC) (h). All the precipitation index changes show significant correlations with temperature increment.

718