# Peer review of "Variations of future precipitations in Poyang Lake Watershed under the global warming using a spatiotemporally distributed downscaling model"

_Hydrology and Earth System Sciences, 2018_

## Referee Comment (RC1) · X.Y. Meng (Referee) · 17 Jul 2018

General comment

This manuscripts developed a spatiotemporally distributed downscaling method and analyzed the precipitation changes under climate warming. The method was applied in the Poyang Lake watershed. Climate warming is a hot topic and the spatial downscaling method is interesting. I do, however, have a number of suggestions and questions that needs to be addressed clearly before it can be published.

Special comment

1. The authors use MRI-CGCM3 data to estimate the future precipitation changes. Why do you choose MRI-CGCM3 data, not other Global Climate Models?

2. The authors use precipitation simulations in RCP8.5 scenario form MRI-CGCMs to estimate precipitation changes under future climate warming. Why do you choose only RCP8.5 scenario, instead of other scenarios?

3. The authors analyze the future precipitation changes in the Poyang Lake watershed using a Global Climate Model. The Poyang Lake watershed is a small area; while the Global Climate Model is coarse with resolution larger than $1° \times 1°$, which is difficult to be applied in a local scale such as the Poyang Lake watershed. The application could be reconsidered.

4. In the methodology section, there is some confusions. What is the relationship between the STDDM and linear-scale algorithm? That should be explained more clearly.

5. By STDDM, you calculate the precipitation of each grid separately and get the downscaled precipitations. The downscaled precipitation is grid data. There may be some outstanding grid in which the precipitation is far different from the adjacent grids. According to first law of geography, near things are more related than distant things. So I suggest that the downscaled precipitation should be smoothed by smoothing filter.

6. In 4.1 section, the validation period is from 1986 to 2005. However the observation data is from 1961 to 2005. Why not validate the downscaled precipitation in the same period from 1961 to 2005?

7. Line199: The sentence missed a comma.

8. There are 69 references. Please provide the reference number for each reference. Is every reference useful to the research? If not, please delete some.

9. Line197-200: Monthly precipitations, > 75% percentile of the 12 monthly precipitations, were classified as the extreme wet monthly precipitations for each year of the 103 years; monthly precipitations, $\leq 25\%$ percentile were classified as the extreme dry monthly precipitation. The monthly precipitation of 25%-50% and 50%-75% quantiles are classified as normal dry and wet monthly precipitations. Why do the author classify the monthly precipitation into 4 categories, not 5 or 7? Why choose 25%, 50%, 50% and 75% quantiles as the classified boundary?

---

## Referee Comment (RC2) · Anonymous Referee #2 · 22 Oct 2018

Quantifying the spatio-temporal heterogeneities in the downscaling process is of great importance for future projections of climate change, especially for regional water resources adaptive management. The spatiotemporally distributed downscaling model (STDDM) proposed in this manuscript is of certain significance for statistical downscaling methodology, which could improve the simulation effects of climate variables at different spatial and temporal scales. However, the results, conclusions and discussion presented in current manuscript is not clear, concise, and well structured. The manuscript in current form needs a major revision before acceptance. Specific comments are as follows:

1. Assemble projection based on multi-GCMs has been widely used for regional future climate change scenarios, which is referred as the mainstream and popular method in the downscaling technique. However, only one GCM MRI-CGCM3 was selected in this study, based on the conclusions from Yuan et al. (2013) indicating a better performance in simulating diurnal rainfall over subtropical China, which is not enough for performance evaluation of multi-GCMs from CMIP5 in the specific Poyang Lake basin.

2. In order to detect sensitivity of precipitation change under global climate warming, different RCP scenarios should be selected to do comparative analysis. However, only RCP 8.5 was selected to generate future climate change scenarios in current manuscript, which is insufficient to obtain a scientific and convinced projection for the study area.

3. Too many time periods are defined in the manuscript corresponding to different years, such as baseline and future periods, historical, historical extent and future, etc., which would make readers confused and difficult to understand.

4. It will be better to add an evaluation section for the gridded meteorological data by using gauging stations observation.

5. English writing is poor in current manuscript, which needs to be polished by a native English-speaking editor. Examples of grammar errors are as follows:

Line 27: threating to → threatening

Line 37: constructed → constructs

Line 43: in the station scale → at the station scale, many similar errors in other paragraphs.

Line 45: as underlays of local region is complex → as underlays of local region are

complex

Line 57: project → projects

Line 69: Precipitation redistributions under global warming has → Precipitation redistributions under global warming have

Line 77: includes → include

Line 84: metrological → meteorological, many similar errors in other sentences.

Figure 2, 1(a): observitions → observations

———————————————

---

## Author Response (AR1)

**Reply to Referee Comment 1**

We are very grateful to the reviewer for reading the manuscript extremely carefully and forwarding the valuable suggestions for improvement. Point-by-point responses to the reviewers' comments are listed below.

**The reviewer's comment 1:** The authors use MRI-CGCM3 data to estimate the future precipitation changes. Why do you choose MRI-CGCM3 data, not other Global Climate Models?

**Authors' response:** Thank you very much for the suggestions.

Compared to the other CGMs of CMIP5, the MRI-CGCM3 (Meteorological Research Institute Coupled Ocean-Atmosphere General Circulation Model3) performs better in simulating diurnal rainfall over subtropical China (Yuan et al. 2013) and has the finest resolution of 1.121° × 1.125°, thus being applied in Poyang Lake Watershed. And MRI-CGCM model is just a study case to examine the performance of STDDM. Other single-model is also ok to test the applicability of STDDM.

**The references:**

Yuan, W.: Diurnal cycles of precipitation over subtropical China in IPCC AR5 AMIP simulations, Adv. Atmos. Sci., 30(6), 1679–1694, doi:10.1007/s00376-013-2250-9, 2013.

**The related content is in the manuscript in L114-120.**

**The reviewer's comment 2:** The authors use precipitation simulations in RCP8.5 scenario from MRI-CGCMs to estimate precipitation changes under future climate warming. Why do you choose only RCP8.5 scenario, instead of other scenarios?

**Authors' response:** Thank you very much for the suggestions.

The future data includes simulations of the Representative Concentration Pathways (RCPs) of 8.5, 6.0, 4.5 and 2.6. Compared to the other RCPs, temperature increases the most in the RCP8.5 scenario, which corresponds to a highest greenhouse gas emission, leading to a radiative forcing of 8.5 W/m2 and temperature increment of 7.14 °C at the end of 21st century.

The research is to detect obvious changes of precipitations under climate warming. What we should do is

to display the significant change of precipitations in a scenario where temperature increment is large enough. Precipitation changes can be detected the most obviously under the climate warming scenario with temperature increasing the most. Compared to the other RCPs, the temperature in RCP8.5 scenario increased the most. So we select future simulations in the RCP8.5 scenario.

**The related content is in L.121-125.**

**The reviewer's comment 3:** The authors analyze the future precipitation changes in the Poyang Lake watershed using a Global Climate Model. The Poyang Lake watershed is a small area; while the Global Climate Model is coarse with resolution larger than 1 °x 1 °, which is difficult to be applied in a local scale such as the Poyang Lake watershed. The application could be reconsidered.

**Authors' response:** Thank you very much for the suggestions.

The Poyang Lake watershed is one of the major grain producing areas of China. In the south of the watershed, there is an internationally important habitat for migratory birds, abundant of biodiversity and regarded as Natural Reserve. The watershed is also a vital part of Yangtze River Economic Belt. However, floods and droughts occurs fluently in the Poyang Lake watershed, which cannot be immune to climate warming. As an important economic and ecological zones, what the precipitations changes in spatiotemporal distribution will be under the climate worming is a concern.

GCMs is a basic tool to analyze the future climate changes. As the resolution of GCMs is coarse unable to applied in small scale such as Poyang Lake Watershed, we downscaled the climate variables in the watershed with resolution of 20 km x 20 km. The uncertainty is $\leq 4.9\%$, demonstrating that the downscaled data can be applied in the Poyang Lake watershed.

**The related content is in L68-76, L29-33 and L254-256 .**

**The reviewer's comment 4:** In the methodology section, there is some confusions. What is the relationship between the STDDM and linear-scale algorithm? That should be explained more clearly.

**Authors' response:** Thank you very much for the suggestions.

STDDM is a logical frame, including three parts: upsampling GCMs simulations, constructing mapping relationships between the GCMs simulations and local observations, and correcting the GCMs

simulations. In the part 2 constructing relations, a transform function were built between the simulations and the local observations to transform simulations to no-bias data. The transform function could be any bias corrected model, including linear scaling, local intensity scaling, power transformation, distribution mapping models (Teutschbein et al. 2012) and so on. The transform model can be linear or no-linear regressions model. That is the relationship between the simulations and observations. In the study, the linear scaling algorithm was used as a transform function (also called as bias-corrected model), as a case study.

**The references:**

Teutschbein C, Seibert J. Bias correction of regional climate model simulations for hydrological climate-change impact studies: Review and evaluation of different methods [J]. Journal of Hydrology, 2012, 456: 12-29.

**The revised paragraph of manuscript (Line 166-167):**

**Before the revises:**

The bias correction was processed by using the translation function between match-ups of the up-sampled simulation and observation, which is the relations of the match-ups.

**After the revises:**

The bias correction was processed by the transform function between match-ups of the upsampled simulation and observations, which represents the mapping relationship between the match-ups. The transform function could be any bias corrected model, including linear scaling, local intensity scaling, power transformation, distribution mapping models (Teutschbein et al. 2012) and other linear or nonlinear regression models.

**The reviewer's comment 5:** By STDDM, you calculate the precipitation of each grid separately and get the downscaled precipitations. The downscaled precipitation is grid data. There may be some outstanding grid in which the precipitation is far different from the adjacent grids. According to first law of geography, near things are more related than distant things. So I suggest that the downscaled precipitation should be smoothed by smoothing filter.

**Authors' response:** Thank you very much for the suggestions.

The downscaled climate data is calculated based on the relationships between the up-sampled simulations and observations. The up-sampled simulations and observations are grid data. The relationships are the transform function between the match-ups of the simulation and observation. The transform function is constructed separately for match-ups in different grids. The grid data, including the simulations and observations, follows the first law of geography that the climate variable value is more related than distant grids. So the transform function based on the match-ups in nearer grids is more related than distant grids. Consequently, the climate variables calculated by the transform function should also follows the first low of geography. Besides, the downscaled results (precipitations in Fig. 9) shows almost no outstanding grid, which demonstrates that the results follows the first low of geography.

On the contrary, smoothing may lead to information missing of the climate variables.

So I think there could be no need to do smoothing.

**The reviewer's comment 6:** In 4.1 section, the validation period is from 1986 to 2005. However the observation data is from 1961 to 2005. Why not validate the downscaled precipitation in the same period from 1961 to 2005?

**Authors' response:** Thank you very much for the suggestions.

To avoid model overfitting, there should be calibrations and validations. In the study, the calibration and validation periods are from 1961 to 1985 and 1986 to 2005, separately. The downscaled model is constructed based on the data in calibration period. We should also need to know whether the model could be applied in the data of different time. So the validation period is different from the calibrations.

The model could be more correctly base on more data. So at last, we used all data from 1961 to 2005 to reconstruct the downscaling model.

**The related content is in Line 170-172 and Line 227-230.**

**The reviewer's comment 7:** Line199: The sentence missed a comma.

**Authors' response:** Thank you very much for the suggestions.

It has been revised in the manuscript.

**The reviewer's comment 8:** There are 69 references. Please provide the reference number for each reference. Is every reference useful to the research? If not, please delete some.

**Authors' response:** Thank you very much for the suggestions.

There is no need to add references number in the manuscript. All the references are useful to the study.

**The reviewer's comment 9:** Line197-200: Monthly precipitations, > 75% percentile of the 12 monthly precipitations, were classified as the extreme wet monthly precipitations for each year of the 103 years; monthly precipitations, ≤ 25% percentile were classified as the extreme dry monthly precipitation. The monthly precipitation of 25%-50% and 50%-75% quantiles are classified as normal dry and wet monthly precipitations. Why do the author classify the monthly precipitation into 4 categories, not 5 or 7? Why choose 25%, 50%, 50% and 75% quantiles as the classified boundary?

**Authors' response:** Thank you very much for the suggestions.

As the extreme wet and dry months cause floods and droughts more frequently, we pay more attention to the precipitations changes in extreme wet or dry months. So the months are differentiated as extreme and normal ones. The precipitation changes in wet and dry months also could show different condition, so the precipitation months in dry and wet should be separated. Here, we differentiate the months as the extreme wet, extreme dry, normal wet and normal dry ones. As for the classified boundary, it is more flexible. Several tries showed that 25%, 50%, 50% and 75% quantiles is appropriate classified standard. However, other classified standard is also OK, only if the precipitation changes of extreme wet, extreme dry, normal wet and normal dry months could be differentiated.

**Reply to Referee Comment 2**

We are very grateful to the reviewer for reading the manuscript extremely carefully and forwarding the valuable suggestions for improvement. Point-by-point responses to the reviewers' comments are listed below.

**1. General comments**

**Reviewer's comment:** However, the results, conclusions, and discussion presented in the current manuscript are not clear, concise, and well structured.

**Authors' response:** Thank you very much for the suggestion.

As you suggested, the results, conclusions, and discussion have been revised in the manuscript.

**2. Specific Comments**

**Reviewer's comment 1:** Assemble projection based on multi-GCMs has been widely used for regional future climate change scenarios, which is referred as the mainstream and popular method in the downscaling technique. However, only one GCM MRI-CGCM3 was selected in this study, based on the conclusions from Yuan et al. (2013) indicating a better performance in simulating diurnal rainfall over subtropical China, which is not enough for performance evaluation of multi-GCMs from CMIP5 in the specific Poyang Lake basin.

**Authors' response:** Thank you very much for the suggestion.

The research is mainly aimed to propose a spatiotemporal distributed downscaling method which could be applied to every single-GCM model. The MRI-CGCM3 is a study case to examine the model performance or availability of STDDM, as well as Poyang Lake which is taken as a study area. The validation is operated in several aspects. For the historical data, simulations from the GCM-downscaled result by STDDM and observations from meteorological stations were compared (Section 4.1). For the future data, we compared the future period (2081-2100) with the baseline period (1998-2017). The intra-annual and inner-annual variability were analyzed. The precipitation changes were also explained by climate warming in section 4.4. The explanation suggests the downscaling method is reasonable and STTDM could be applied in the basin-scale region based on a GCM successfully. The examination on a test GCM is necessary before STDDM could be used in other GCMs or multi-GCMs.

Indeed assemble projection is a mean stream. The model ensemble has a better model performance than the single-model assessed by R (correlation coefficient) and RMSE (Root-Mean-Square Error). However, the model ensemble is burdened with a smaller standard deviation (SD) than the single-models and observations (Fig 1). Except for R and RMSE, SD is also an important model evaluation index. The small SD value means small fluctuation, which demonstrates the fluctuation signal of original models (the signal-models) is not kept completely after being assembled. The SD of multi-GCMs is usually smaller

than original models (Fig 1). The monthly and daily variation are weakened in model ensembles. The ensemble can hardly analyze the seasonal or daily extreme event change exactly, taking the extreme dry (or wet) months and the max daily precipitation for example. In the study, we analyze the seasonal variations, as well as the change of extreme event intensities and frequencies. Using multi-GCMs can hardly reflect the application accuracy of STDDM precisely, in extreme climate analysis based on monthly and daily data. So a single-model should be selected to test the performance of STDDM.

A specific single-model could be used to analyze the seasonal change exactly, especially the extreme climate event change. As MRI-CGCM3 has the best spatial resolution among the CMIP5 GCMs, and a better performance in simulating diurnal rainfall over subtropical China, we took MRI-CGCM3 as a test case to apply in the Poyang Lake Basin and examine whether the STDDM can be used to produce reasonable monthly and daily data, especially the extreme climate change.

The title *Variations of future precipitations in Poyang Lake Watershed under global warming using a spatiotemporally distributed downscaling model* might confuse you. So it will be revised as *Precipitation projection using a spatiotemporal distributed method: a case study in Poyang Lake Basin based on MRI-CGCM3*. And the content will be revised corresponding to the revised title.

In summary, using multi-GCMs instead of MRI-CGCM3 in the study could be reconsidered.

[Figure]

Figure 1. The Taylor figures (Taylor et al.,2001) of model evaluation. Following Taylor et al. (2001), the radial distance from the origin denotes the standard deviation of each data set (the primary observations are shown as a red line) and the angular distance from the horizontal denotes the correlation coefficient r between each model data set and the primary observations. The centered RMS error (RMSE') is indicated by the distance to the intersection of the green dashed line and the horizontal axis with units and magnitude indicated by the radial axis. The model ensemble is constructed by a genetic algorithm. The genetic algorithm is used to calculate the best weigh for each single-model, assuming RMSE' as the cost function. The model ensemble is the weighted sum of each single-model. The other single-models include ACCESS1-0, ACCESS1-3, BCC-CSM1-1-m, BCC-CSM1-1, BNU-ESM, CanESM2, CCSM4, CMCC-

CMS, CMCC-CM, CNRM-CM5, FGOALS-g2, GISS-E2-H-CC, GISS-E2-H, GISS-E2-R-CC, GISS-E2-R, HadGEM2-AO, HadGEM2-CC, HadGEM2-ES, inmcm4, IPSL-CM5A-LR, IPSL-CM5A-MR, IPSL-CM5B-LR, MIROC-ESM-CHEM, MIROC-ESM, MIROC5, MPI-ESM-LR, MPI-ESM-MR, MRI-CGCM3 and NorESM1-M. The model description can be obtained from *https://pcmdi.llnl.gov/mips/cmip5/availability.html*.

**The references:**

Taylor K E. Summarizing multiple aspects of model performance in a single diagram [J]. Journal of Geophysical Research: Atmospheres, 2001, 106(D7): 7183-7192.

**The main revised paragraph of manuscript (Line 1-2):**

**Before the revises:**

Variations of future precipitations in Poyang Lake Watershed under the global warming using a spatiotemporally distributed downscaling model

**After the revises:**

Precipitation projections using a spatiotemporal distributed method: a case study in the Poyang Lake Watershed based on MRI-CGCM3

**Reviewer's comment 2:** In order to detect the sensitivity of precipitation change under global climate warming, different RCP scenarios should be selected to do comparative analysis. However, only RCP 8.5 was selected to generate future climate change scenarios in the current manuscript, which is insufficient to obtain a scientific and convinced projection for the study area.

**Authors' response:** Thank you very much for the suggestion.

The future data includes simulations of the Representative Concentration Pathways (RCPs) of 8.5, 6.0, 4.5 and 2.6. Compared to the other RCPs, temperature increases the most in the RCP8.5 scenario, which corresponds to a highest greenhouse gas emission, leading to a radiative forcing of 8.5 W/m2 and temperature increase of 7.14 °C at the end of 21st century.

The research is to detect remarkable precipitation changes under climate warming, which should be pronounced enough to be acknowledged by us. To get the obvious precipitation changes, what we should do is to obtain the future precipitation in a high-emission scenario where the temperature increment is large enough. Compared to the other RCPs, the temperature increment in RCP8.5 scenario is the largest. So we select future simulations in the RCP8.5 scenario.

Although it is valuable to detect the sensitivity of precipitation change, the sensitivity analysis is not the purpose of the study. And there are many climate change related researches (Gourdji et al.,2013; Sillmann et al.,2013; De et al., 2014; Cai et al.,2017) only use the high-emissions scenario to investigate the impacts of climate warming. The result from RCP8.5 scenario is the most remarkable, from which we can get the obvious change and know what will happen when climate warming gets worse. The study is to investigate the remarkable change of precipitation under climate warming.

So it could be reasonable to only select RCP85 scenario in the experiment to detect the significant changes of precipitation.

**The references:**

De Lavergne C, Palter J B, Galbraith E D, et al. Cessation of deep convection in the open Southern Ocean under anthropogenic climate change [J]. Nature Climate Change, 2014, 4(4): 278.

Cai W, Li K, Liao H, et al. Weather conditions conducive to Beijing severe haze more frequent under climate change[J]. Nature Climate Change, 2017, 7(4): 257.

Gourdji, S. M., Sibley, A. M. & Lobell, D. B. Global crop exposure to critical high temperatures in the reproductive period: Historical trends and future projections. Environ. Res. Lett. 8, 024041 (2013).

Sillmann J, Kharin V V, Zwiers F W, et al. Climate extremes indices in the CMIP5 multi-model ensemble: Part 2. Future climate projections[J]. Journal of Geophysical Research: Atmospheres, 2013, 118(6): 2473-2493.

**The main revised in the manuscript (Line 122-123):**

**Before the revises:**

Thus, to detect more sensitive precipitation change under climate warming, we selected future simulations in the RCP8.5 scenario.

**After the revises:**

The research is to detect the remarkable precipitation changes under climate warming; thus we selected future simulations in the RCP8.5 scenario.

**Reviewer's comment 3:** Too many time periods are defined in the manuscript corresponding to different years, such as baseline and future periods, historical, historical extent and future, etc., which would make readers confused and difficult to understand.

**Authors' response:** Thank you very much for the suggestion.

Cmip5 GCMs include historical (1850-2005), historical extent (2006-2012), RCPs (2005-2100 or 2005-2300) scenarios (Friedlingstein et al., 2008). At the WGCM meeting in October 2011, there was agreement that it would be useful to extend the CMIP5 historical runs to near-present 2012, rather than ending them in 2005 (Friedlingstein et al., 2008). So another scenario (historical extension) from 2006 to 2012 was constructed to extend the historical data to 2012. In the study, we merge the historical (from 1961 to 2005), historical extent (from 2006 to 2012) and RCP85 (from 2013 to 2100) data, as merged data (1961-2100). From the merged data, simulations from 1998 to 2017 were selected as the baseline period data, and simulations from 2081 to 2100 were selected as the future period data.

**The references:**

Friedlingstein OB, Webb M, Gregory J. A Summary of the CMIP5 Experiment Design [J]. 2008.

**The main revised paragraph of manuscript (Line 117-134):**

**Before the revises:**

From MRI-CGCM3, we select historical (1961 to 2005), historical extent (2006 to 2012) and future (2006 to 2100) precipitation and temperature simulations. The future data includes simulations of the Representative Concentration Pathways (RCPs) of 8.5,6, 4.5 and 2.6. Compared to the other RCPs, in the RCP8.5 scenario temperature increases the most, which is corresponds to a highest greenhouse gas emission, leading to a radiative forcing of 8.5 W/m2 and temperature increase of 7.14 °C at the end of 21st century (Taylor et al. 2012). Thus, to detect more sensitive precipitation change under climate warming, we selected future simulations in the RCP8.5 scenario.

The local grid observations (Zhao et al., 2014) with a resolution of 0.5°×0.5° are downloaded from the China Meteorological Data Service Center (http://data.cma.cn/). The local grid observations and MRI-CGCM3 historical simulations were used to construct relationship to correct the MRI-CGCM3 data. China metrology point data were also downscaled and used to validate the bias-corrected MRI-CGCM3 simulations. To investigate the relationship between precipitation changes and the temperature increase, we extract not only temperature data, but also precipitations.

To quantitatively analyse the precipitation changes under climate warming in 21st century, we compared precipitation between the baseline and future period. As annual precipitation observations have main oscillation periods of quasi-20 years (Zhan et al. 2011), we selected three 20 years, the baseline period from 1998 to 2017, the near future period from 2041 to 2060 and the far future period from 2081 to 2100. We merge historical simulations from 1998 to 2005, and historical extent simulations from 2006 to 2012, and RCP8.5 simulations from 2013 to 2017, which is the nearest 20 years and thus selected as the baseline period. The data in near and far future period are derived from simulations in RCP8.5 scenarios.

**After the revises:**

Thus we select MRI-CGCM3 data applied in Poyang Lake Watershed to test the performance of the STDDM.

The future data of MRI-CGCM3 includes simulations of the Representative Concentration Pathways (RCPs) of 8.5,6, 4.5 and 2.6. Compared to the other RCPs, in the RCP8.5 scenario temperature increases the most, which is corresponds to a highest greenhouse gas emission, leading to a radiative forcing of 8.5 W/m2 and temperature increase of 7.14 °C at the end of 21st century (Taylor et al. 2012). The research is to detect the remarkable precipitation changes under climate warming; thus we selected future simulations in the RCP8.5 scenario. In the study, we merge the historical (from 1961 to 2005), historical extent (from 2006 to 2012) and RCP85 (from 2013 to 2100) data, as the merged data (1961-2100). To quantitatively analyze the precipitation changes under climate warming in 21st century, we compared precipitation between the baseline and future period. As annual precipitation observations have main oscillation periods of quasi-20 years (Zhan et al. 2011), we selected three 20 years from the merged data. From the merged data, simulations from 1998 to 2017 were selected as the baseline period data, simulations from 2041 to

 2060 were selected as the near future period data, and simulations from 2081 to 2100 were selected as the

172 further future period data.

173 The local grid observations (Zhao et al., 2014) with a resolution of 0.5°×0.5° are downloaded from the

174 China Meteorological Data Service Center (http://data.cma.cn/). The local grid observations and MRI-

175 CGCM3 historical simulations were used to construct relationship to correct the GCM data. China

176 metrology point data were also downscaled and used to validate the grid observations and the downscaled

177 GCM simulations. To investigate the relationship between precipitation changes and the temperature

178 increment, we extract not only precipitations, but also temperature.

179

180 **Reviewer's comment 4:** It will be better to add an evaluation section for the gridded meteorological data

181 by using gauging stations observation.

182 **Authors' response:** Thank you very much for the suggestion.

183 The evaluation for the gridded meteorological data has be added in the manuscript.

184 **The following was added in the manuscript (Line 238-239):**

185 Validation about the China meteorological grid observations should be performed, as well as the STDDM.

186 As the STDDM introduce the China meteorological grid observations and the grid data is not the direct

187 in-suit data, validation about the gridded data is necessary. The determination coefficient (R2), root mean

188 square error (RMSE) and PBias (percent bias) were used to examine the model performance.

189 **The following was added after Line 570:**

190

[Figure]

191

192 Fig. 3. Validation of gridded meteorological data (GridObs) by using gauging stations observation:

193 Precipitation (pcp; a,b,c and d) and temperature (tem; e,d,f and g) at meteorological station of Jian (a and

194 e), Ganzhou (b and d), Zhangshu (c and f) and Lushan (d and g).

195

196 **Reviewer's comment 5:** English writing is poor in the current manuscript, which needs to be polished

by a native English-speaking editor. Examples of grammar errors are as follows:

Line 27: threating to → threatening

Line 37: constructed → constructs

Line 43: in the station scale → at the station scale, many similar errors in other paragraphs.

Line 45: as underlays of the local region is complex → as underlays of local region are complex

Line 57: project → projects

Line 69: Precipitation redistributions under global warming has → Precipitation redistributions under global warming have

Line 77: includes → include

Line 84: metrological → meteorological, many similar errors in other sentences. Figure 2, 1(a): observitions → observations

**Authors' response:** Thank you very much for the language editing.

The writing errors has been revised in the manuscript.

[revised manuscript text omitted]

---

## Author Response (AR2)

**Comments from the editor**

We are very grateful to the editor for reading the manuscript extremely carefully and forwarding the valuable suggestions for improvement. Point-by-point responses to the editor's comments are listed below.

**The editor's comment 1:** There were published not a few downscaling algorithms keeping spatial correlation structure of meteorological fields (Charles et al., 1999; Mohamed Ali et al., 2017 among many others). The corresponding studies should be mentioned, and advantages of the algorithm developed in this study should be clearly denoted in Introduction.

Also, I recommend remove the first sentence ("Traditional statistical downscaling methods are performed on independent station measurements and ignore spatial correlations and spatiotemporal heterogeneity") from the Abstract.

Ben Alaya, Mohamed Ali & Ouarda, Taha & Chebana, Fateh. (2017). Non-Gaussian spatiotemporal simulation of multisite daily precipitation: downscaling framework. Climate Dynamics. 10.1007/s00382-017-3578-0.

Charles, Stephen & Bates, Bryson & P. Hughes, James. (1999). A spatiotemporal model for downscaling precipitation occurrence and amounts. Journal of Geophysical Research. 104. 31657-31669. 10.1029/1999JD900119.

**Authors' response:** Thank you very much for the suggestions.

The main advantage of the method developed (STDDM) is reproducing downscaled data in the spatially continuous grid scale, instead of the discrete station scale. Compared to the discrete points, the spatial continuous grid data can express the spatial distribution of climate variables more accurately and clearly. In addition, the method (STDDM) also consider the spatial and temporal heterogeneity of the climate variables, as the factors driving the climate variable vary in space and time. For different space and different time, the specific algorithm or parameter of the downscaling method is different.

In the old version of the manuscript, the sentence '*traditional statistical downscaling methods … ignore spatial correlations*' may be not accurate. The main idea here is as the following.

The classic downscaling method (Charles et al., 1999; Mohamed Ali et al., 2017 for example) was performed on discrete points, and produced downscaled data in the discrete station scale. Compared to the discrete points, the spatial continuous grid data can express the spatial distribution of climate variables more accurately and clearly. Consequently, the spatial continuous grid data can express the spatial correlation and heterogeneity more accurately and clearly. Additionally, the method (STDDM) also consider the spatial and temporal heterogeneity of the climate variables. For different space and different time, the specific algorithm or parameter of the downscaling method is different.

**The revised in the manuscript (Line 42-68):**

**Before the revises:**

[revised manuscript text omitted]

**The editor's comment 2:** Page 3; line 52. Strictly saying, only MIKE SHE is a distributed model. Two others are semi-distributed ones. Please change

**Authors' response:** Thank you very much for the suggestions.

**The revised in the manuscript (Line 51-53):**

**Before the revises:**

Additionally, spatially continuous data can be directly used in a spatially distributed hydrological model, such as Crest (Wang et al., 2011), VIC (Lohmann et al. 1998), and MIKE SHE (DHI, 2014), which is the forefront of international hydrological scientific research (Beven et al. 1990).

**After the revises:**

Additionally, spatially continuous data can be directly used in a spatially distributed or semi-distributed hydrological model, such as Crest (Wang et al., 2011), VIC (Lohmann et al. 1998), and MIKE SHE (DHI, 2014), which is the forefront of international hydrological scientific research (Beven et al. 1990).

**The editor's comment 3:** Eq. 4. It is not necessary to give this well-known formula. Also, replace VC with the "coefficient of variation" (CV) throughout the text

**Authors' response:** Thank you very much. It was revised in the manuscript.

**The main revised in the manuscript (Line 214-226):**

**Before the revises:**

[revised manuscript text omitted]